# Identification of a HTT-specific binding motif in DNAJB1 essential for suppression and disaggregation of HTT

S. M. Ayala Mariscal[1], M. L. Pigazzini[1,2], Y. Richter[3], M. Özel[3], I. L. Grothaus [4,5], J. Protze [1], K. Ziege[1], M. Kulke [6], M. ElBediwi [3], J. V. Vermaas [6], L. Colombi Ciacchi[4,5,7], S. Köppen [4,7], F. Liu [1] & J. Kirstein [1,3] ✉

Huntington's disease is a neurodegenerative disease caused by an expanded polyQ stretch within Huntingtin (HTT) that renders the protein aggregation-prone, ultimately resulting in the formation of amyloid fibrils. A trimeric chaperone complex composed of Hsc70, DNAJB1 and Apg2 can suppress and reverse the aggregation of HTTExon1Q$_{48}$. DNAJB1 is the rate-limiting chaperone and we have here identified and characterized the binding interface between DNAJB1 and HTTExon1Q$_{48}$. DNAJB1 exhibits a HTT binding motif (HBM) in the hinge region between C-terminal domains (CTD) I and II and binds to the polyQ-adjacent proline rich domain (PRD) of soluble as well as aggregated HTT. The PRD of HTT represents an additional binding site for chaperones. Mutation of the highly conserved H244 of the HBM of DNAJB1 completely abrogates the suppression and disaggregation of HTT fibrils by the trimeric chaperone complex. Notably, this mutation does not affect the binding and remodeling of any other protein substrate, suggesting that the HBM of DNAJB1 is a specific interaction site for HTT. Overexpression of wt DNAJB1, but not of DNAJB1$^{H244A}$ can prevent the accumulation of HTTExon1Q$_{97}$ aggregates in HEK293 cells, thus validating the biological significance of the HBM within DNAJB1.

Huntington's disease (HD) is a neurodegenerative disorder caused by the expansion of a glutamine stretch in the first exon of the protein huntingtin (HTT)[1]. The disease pathology is fully penetrant above the threshold of Q ≥ 39 with an inverse correlation between the polyQ length and the age of onset and severity of the disorder. HTT is a large ubiquitous protein of 3144 amino acids. Alternative splicing and several caspase-mediated cleavage events reduce the full-length HTT into progressively smaller fragments. The N-terminal first exon of HTT

(HTTExon1) contains the pathological polyQ expansion[2]. HTTExon1 with an expanded polyQ stretch is responsible for the formation of amyloid fibrils found in both the cytoplasm and nucleus of HD patient's neurons[3,4]. HTTExon1Q$_n$ forms amyloid fibrils in vivo and in vitro[5,6]. Amyloid formation is facilitated by the polyQ domain, that is flanked by an N-terminal stretch of 17 amino acids (N17) and a C-terminal proline-rich domain (PRD). The highly conserved N17 enhances the aggregation potential of HTT[7]. The evolutionary newer PRD is

[1]Leibniz Research Institute for Molecular Pharmacology (FMP) im Forschungsverbund Berlin e.V. (FMP), Berlin, Germany. [2]NeuroCure Cluster of Excellence, Charité Universitätsmedizin Berlin, Berlin, Germany. [3]Department of Cell Biology, University of Bremen, Bremen, Germany. [4]Hybrid Materials Interfaces Group, Faculty of Production Engineering and Bremen Center for Computational Materials Science, University of Bremen, Bremen, Germany. [5]Center for Environmental Research and Sustainable Technology (UFT), University of Bremen, Bremen, Germany. [6]MSU-DOE-Plant Research Laboratory, Michigan State University, 612 Wilson Road, East Lansing, MI 48824, USA. [7]MAPEX Bremen Center for Materials and Processes, University of Bremen, Bremen, Germany. ✉e-mail: kirstein@uni-bremen.de

composed of two stretches of proline repeats (P1 and P2) separated by a proline/glutamine-rich region[8]. Several studies could assign a protective role to the PRD as it reduced or delayed HTT aggregation[8–10].

The aggregation propensity of HTT is regulated by molecular chaperones, and the modulation of expression for Hsp70, J-domain proteins (JDPs), Hsp110, NAC, TRiC, and MOAG-4/SERF have been demonstrated to reduce HTT/polyQ aggregation and toxicity in various model systems[11–19]. Yet, a complete suppression of HTTExon1Q$_{48}$ aggregation could only be shown for the trimeric chaperone complex, composed of Hsc70, a class II J-domain protein (JDP), and the nucleotide exchange factor Apg2[18]. Hsc70, Apg2, and DNAJB1 can suppress the amyloid fibril formation but also resolubilize HTTExon1Q$_{48}$ fibrils[18], suggesting a shared binding epitope between soluble and aggregated HTT.

We set out to map the binding interface between the trimeric chaperone complex and HTTExon1Q$_{48}$ to gain mechanistic insights into the dual mode of action of this chaperone complex: suppressing HTT aggregation and disaggregating preformed HTT amyloid fibrils. Using cross-linking mass spectrometry, we could detect the interaction between DNAJB1/Hsc70 and HTTExon1Q$_{48}$. We could define an HTT-binding motif (HBM) within DNAJB1 that is specific for the remodeling of HTT. The nine amino acid stretch is located in the C-terminal domain of DNAJB1 and interacts with the second polyproline stretch of the PRD of HTTExon1Q$_{48}$. Mutation of the highly conserved H244 within the HBM completely abrogated the suppression and disaggregation of HTTExon1Q$_{48}$ fibrils by the chaperone complex. We could validate the biological significance of H244 in HEK293 cells that overexpress HTTExon1Q$_{97}$. Overexpression of wild-type (wt) DNAJB1$^{wt}$, but not of DNAJB1$^{H244A}$ could suppress the aggregation of HTTExon1Q$_{97}$. Molecular dynamics simulations revealed that H244 is crucial to stabilize the binding of DNAJB1 to HTTExon1Q$_{48}$, and that mutation of this site prevents the formation of a stable complex. The second proline stretch within the HTT PRD represents an additional binding site for chaperones and is accessible in both the soluble and fibrillar states of HTT. This shared binding site in soluble and aggregated HTT could explain why the same chaperone complex can suppress the aggregation of still soluble, monomeric HTT and also resolubilizes HTT fibrils. We provide further evidence that in addition to the HBM of DNAJB1, the highly flexible G/F-rich region contributes to the suppression of aggregation. A substitution of the G/F-rich region by that of DNAJB1 enabled the class A J-domain protein DNAJA1 that is otherwise unable to suppress HTTExon1Q$_{48}$ aggregation, to significantly delay the aggregation of HTTExon1Q$_{48}$ together with Hsc70 and Apg2.

## Results

### DNAJB1 binds HTTExon1Q$_{48}$ with its hinge region between CTDI and CTDII

The trimeric chaperone complex Hsc70, DNAJB1, and Apg2 can suppress the aggregation of soluble HTTExon1Q$_{48}$ and is also able to reverse fibrilization by disaggregating HTTExon1Q$_{48}$ amyloid fibrils[18]. Thus, the chaperones interact with soluble as well as aggregated HTTExon1Q$_{48}$. To gain insight into the interaction between HTTExon1Q$_{48}$ and the three chaperones, we employed cross-linking mass spectrometry (XL-MS). To allow for unbiased analysis, we incubated HTTExon1Q$_{23/48}$ with Hsc70, DNAJB1, and Apg2 in the presence of ATP using the non-specific cross-linker succinimidyl 4,4′-azipentanoate (SDA). We also used the non-specific protease elastase in the subsequent digestion step prior to MS analysis. We identified an interaction between HTTExon1Q$_{48}$ and DNAJB1 as well as between HTTExon1Q$_{23}$ and Hsc70, albeit only when DNAJB1 was present, suggesting a shared interaction site when HTT is transferred from DNAJB1 to Hsc70 (Fig. 1a and Supplementary Fig. 1a). Both chaperones bind to the second proline stretch (P2) of the PRD (Fig. 1a, right). This binding interface between HTTExon1 and DNAJB1/Hsc70 adds

an interaction site for chaperones known to bind to HTTExon1 (Fig. 1a, left)[12,14,15]. No interaction could be detected between HTTExon1 and Apg2, which is in agreement with our previous analysis of chaperone binding to HTTExon1Q$_{48}$, where we detected direct interactions between Hsp70 and the J-domain protein, but not between Apg2 and HTTExon1Q$_{48}$ fibrils[18]. The cross-link between HTTExon1Q$_{48}$ and Hsc70 mapped to the nucleotide-binding domain at its N-terminus. This interaction site might be surprising, but recent proteomic analyses demonstrated that Hsp70 interacts with substrates with its nucleotide and substrate binding domains[20]. DNAJB1 binds with its CTD and more precisely with the hinge region between CTDI and CTDII to HTTExon1Q$_{48}$ (Figs. 1a and 4a, c). The CTD of J-domain proteins (JDP) has been previously shown to bind amyloid substrates such as Tau and α-synuclein[21,22]. Here, we could pinpoint the interaction with HTT to 9 amino acids (aa) that are conserved among JDPs from class A and class B (Fig. 1b, green square). Subsequent sequence logo analysis revealed that two positively charged residues, K242 and in particular H244 are highly conserved (Fig. 1b, right). To test the contribution of K242 and H244 to the suppression of HTTExon1Q$_{48}$ fibrilization, we generated point mutations and substituted each residue individually with an alanine. The purity and function of all proteins generated throughout this study were assessed by SDS-PAGE (Supplementary Fig. 2a–c). We employed an established FRET assay to study the fibrilization of HTTExon1Q$_{48}$-CyPet/YPet (Fig. 1c)[18]. The trimeric chaperone complex, Hsc70, DNAJB1 and Apg2 can suppress HTTExon1Q$_{48}$ fibrilization for >20 h in an ATP-dependent manner[18]. Substituting H244 by alanine (H244A) completely abrogated the ability of DNAJB1 to suppress HTT fibrilization together with Hsc70 and Apg2 (Fig. 1d top left, compare magenta curve (DNAJB1$^{wt}$ + Hsc70 and Apg2) with teal curve (DNAJB1$^{H244A}$ + Hsc70 and Apg2)). Mutating K242 to alanine (K242A) strongly decreased the capacity of the trimeric chaperone complex to suppress HTTExon1Q$_{48}$ aggregation, but did not eliminate it (Fig. 1d, top second from left, compare magenta (DNAJB1$^{wt}$ + Hsc70 and Apg2) with the light-blue curve (DNAJB1$^{K242A}$ + Hsc70 and Apg2)). A quantification of the half-life of HTTExon1Q$_{48}$ fibrilization in three independent experiments is depicted on the bottom and demonstrates a significant difference between DNAJB1$^{wt}$ + Hsc70 and Apg2 ($T_{1/2}$ = 27 h) vs DNAJB1$^{H244}$/DNAJB1$^{K242A}$ + Hsc70 and Apg2 ($T_{1/2}$ = 10 h/ 16 h) (Fig. 1d). To test the possibility that the reduced efficiency to suppress HTTExon1Q$_{48}$ fibrilization might be due to a weaker binding of DNAJB1$^{H244A/K242A}$, we doubled the concentration of the respective DNAJB1 variant (while keeping Hsc70 and Apg2 concentrations constant) and did not observe a significant improvement for 2xDNAJB1$^{H244A}$ and only a moderate improvement for 2xDNAJB1$^{K242A}$ in suppression capacity (Fig. 1d, dark-purple and gray curves). We conclude that mutating H244 by alanine severely limits the ability of the trimeric chaperone complex (DNAJB1, Hsc70, and Apg2) to suppress HTTExon1Q$_{48}$ fibrilization, and this defect cannot be rescued by increasing the concentration of the DNAJB1 variant. The contribution of K242 to the binding and suppression of HTTExon1Q$_{48}$ aggregation appears to be less critical.

The positive charge of both conserved residues, K242 and H244, of the 9 aa stretch that we refer to as HTT-binding motif (HBM), raised the question whether a positive charge at these positions is required for the suppression of HTTExon1Q$_{48}$ fibrilization. Indeed, substituting H244 by phenylalanine (H244F) impaired the suppression (Fig. 1d, $T_{1/2}$ = 10 h) as much as DNAJB1$^{H244A}$, whereas substitution by arginine (H244R) improved the capacity for suppression (Fig. 1d, $T_{1/2}$ = 17 h for DNAJB1$^{H244R}$ (light purple) compared to $T_{1/2}$ = 10 h for DNAJB1$^{H244A}$; both conditions with Hsc70 and Apg2; Fig. 1d). A substitution by the polar yet neutral glutamine (H244Q) exhibited a stronger capacity to suppress HTTExon1Q$_{48}$ aggregation than DNAJB1$^{H244A/F/R}$ yet weaker than DNAJB1$^{wt}$ together with Hsc70 and Apg2 (Fig. 1d, $T_{1/2}$ = 24 h for DNAJB1$^{H244Q}$). However, substituting K242 by arginine (K242R) did not

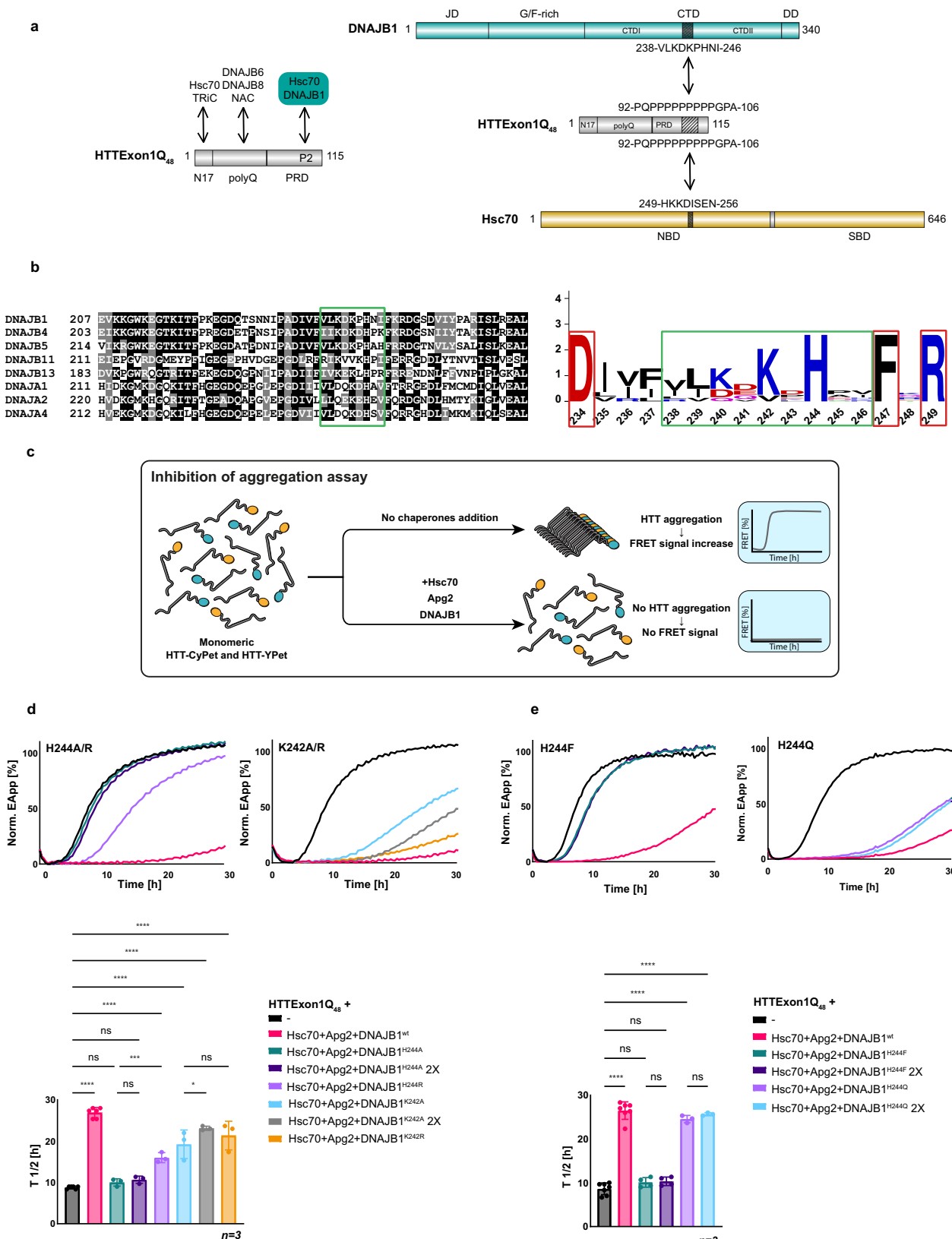

significantly enhance the capacity for suppression of HTTExon1Q48 fibrilization together with Hsc70 and Apg2 (Fig. 1d, orange curve). These data show that either the native histidine, another positive or at least hydrophilic residue (Q) at position 244 within DNAJB1 is strictly necessary for the suppression of HTTExon1Q48 with Hsc70 and Apg2.

The data also shows that substitutions of K242 (either for A or R) were less detrimental.

Based on the identified HBM of the cross-link, we wondered if those 9 aa represent the full binding interface of DNAJB1 or if HBM-adjacent aa also contributes to the binding and hence suppression of

**Fig. 1 | Identification of a HTTExonQ$_{48}$-binding motif within DNAJB1 and Hsc70.**
**a** Left, schematic representation of known binding sites of Hsc70, DNAJB1, DNAJB6/DNAJB8, TRiC, and NAC chaperones on HTTExon1Q$_{48}$. Right, schematic representation of the XL-MS-detected interaction sites between the CTD of DNAJB1/NBD of Hsc70 and the P2 region of the PRD of HTTExon1. **b** Left, alignment of human class A and B J-domain proteins (JDPs) containing an amino acid motif similar to the HTT-binding motif (HBM) of DNAJB1 that is framed in green. Right, the sequence logo of the aligned amino acid of the HBM (green frame) shows the positively charged H244 and K242 residues to be most conserved. The flanking amino acids that are also highly conserved are framed in red. **c** Schematic representation of the FRET-based assay of the analysis of HTTExon1Q$_{48}$ aggregation. In the absence of chaperones, HTTExonQ$_{48}$-CyPet/YPet proteins are in close proximity upon aggregation, which allows energy transfer from excited CyPet to YPet (FRET). The addition of chaperones and ATP prevent HTTExon1Q$_{48}$-CyPet/YPet aggregation and consequently lead to an absence of FRET. **d**, **e** Top, FRET measurements as a readout of HTTExon1Q$_{48}$ aggregation over time in the absence (black curve) and presence of Hsc70, Apg2 and DNAJB1$^{wt}$ (magenta) or variants (**d**) H244A, H244R, K242A, K242R, (**e**) H244F and H244Q. The graphs are representative results of three independent experiments. Bottom, one-way ANOVA analysis of the half-life ($T_{1/2}$) of HTTExon1Q$_{48}$ aggregation of the respective analyses using the same color code as shown on top. Bars represent the mean value and error bars correspond to the mean SD. ****$P \le 0.0001$; ***$P \le 0.001$; *$P \le 0.05$; ns not significant.

HTTExon1Q$_{48}$ aggregation. Using sequence alignments we noticed that in close proximity to the identified 9 aa, three additional aa are 100% conserved, D234, F247, and R249 (Fig. 1b, red squares). We mutated them individually to alanine and subsequently assessed their contribution to the suppression of HTTExon1Q$_{48}$ aggregation. Whereas we could not observe any effect for DNAJB1$^{D234A}$, mutating F247 and R249 to alanine significantly reduced their capacity for suppression of HTTExon1Q$_{48}$ fibrilization with Hsc70 and Apg2 (Fig. 2a). We conclude that F247 and R249 of DNAJB1 are also important residues for the suppression of HTTExon1Q$_{48}$ fibrilization. However, both mutants (DNAJB1$^{F247A}$ and DNAJB1$^{R249A}$) are not as strongly impaired as DNAJB1$^{H244A}$ and hence we focused in the subsequent analyses on H244.

### Mutating H244 of DNAJB1 does not affect the structure and stability of DNAJB1

To exclude that the strong effect of DNAJB1$^{H244A}$ on the suppression of HTTExon1Q$_{48}$ aggregation is due to structural defects inferred by the H244A mutation, we experimentally assessed the stability of the DNAJB1$^{H244A}$ vs DNAJB1$^{wt}$ by three independent assays: size-exclusion chromatography, differential scanning fluorescence (DSF) and circular dichroism (CD) over a temperature range of 20–90 °C. The size-exclusion analysis showed the same elution peak of DNAJB1$^{H244A}$ as DNAJB1$^{wt}$, suggesting that the dimerization of DNAJB1 is not impaired (Supplementary Fig. 1b). The melting curve obtained with DSF is very similar for both mutant and wt protein (Supplementary Fig. 1c), and the CD spectra of DNAJB1$^{H244A}$ and DNAJB1$^{wt}$ are similar as well and only reveal a moderate difference in the melting temperature (DNAJB1$^{wt}$ = 67.78 °C and DNAJB1$^{H244A}$ = 64.74 °C), which is unlikely to affect the activity of the protein at 20–30 °C at which all in vitro assays were performed (Supplementary Fig. 1d). We conclude from these analyses that the H244A mutation is not disrupting the structure and stability of DNAJB1. Accordingly, we also did not observe a defect in activating the ATPase activity of Hsc70 by DNAJB1$^{H244A}$ (Supplementary Fig. 2b). Thus, the interaction with and activation of the partner chaperone Hsc70 is not affected by the H244A mutation of DNAJB1.

### H244 and K242 of DNAJB1 are specific residues required for the binding and remodeling of HTT

Next, we set out to assess the chaperone activity of the trimeric complex with DNAJB1$^{wt}$ vs DNAJB1$^{H244A}$ mutant on different protein substrates. First, we employed the well-established model substrate, firefly luciferase, and studied the refolding activity of heat-denatured luciferase by the trimeric chaperone complex. As depicted, DNAJB1$^{H244A}$ as well as DNAJB1$^{K242A}$ are equally efficient as DNAJB1$^{wt}$ in refolding heat-denatured luciferase together with Hsc70 and Apg2 (Fig. 2b). In all, 40% of the native luciferase could be recovered within 2 h for all chaperone compositions (Fig. 2b). Next, we wondered if H244 and K242 might be only crucial for remodeling amyloid proteins. To test this possibility, we analyzed the aggregation of Aβ$_{1-42}$ in the presence of the trimeric chaperone complex, Hsc70 + Apg2 + either DNAJB1$^{H244A}$, DNAJB1$^{K242A}$ or DNAJB1$^{wt}$. However, mutating K242 and H244 did not affect the suppression of Aβ$_{1-42}$ aggregation either (Fig. 2c). Taken together, these data demonstrate that K242 and in particular H244 of DNAJB1 are specifically required for the suppression of HTTExon1Q$_{48}$ aggregation yet not for every substrate protein.

### DNAJB1 binds to the PRD P2 region of HTTExon1Q$_{48}$

As depicted in Fig. 1a, we obtained crosslinks of Hsc70 and DNAJB1 with the second proline stretch (P2) within the PRD of HTTExon1Q$_{48}$. This binding site was surprising as chaperones have so far only been mapped to either the N17 or the polyQ domain of HTTExon1Q$_n$ (Fig. 1a)[12,14,15]. However, conventional cross-linking approaches used free amino groups such as in lysine that are present only in the N17 domain and are completely absent in the PRD of HTTexon1. Our unbiased strategy of using a non-specific cross-linker and an unspecific protease allowed us to identify the PRD as a binding site for DNAJB1 and Hsc70. To validate this interaction, we generated a HTTExon1Q$_{48}$ variant lacking specifically the P2 domain (HTTExon1Q$_{48}$ΔP2) and tested the ability of the chaperones to suppress the fibrilization of HTTExon1Q$_{48}$ΔP2 (Fig. 3a, b). HTTExon1Q$_{48}$ΔP2 yields a higher plateau in the FRET-based fibrilization assay compared to the non-variated HTTExon1Q$_{48}$, yet the onset of aggregation is not affected. Importantly, the presence of the chaperones, Hsc70, Apg2 and DNAJB1 can no longer suppress HTTExon1Q$_{48}$ΔP2 aggregation (Fig. 3b, compare teal and dark-purple curves) presumably due to the absence of the binding interface (P2). Of note, the slope of the fibrilization curve is also not affected, supporting our assumption that the interaction between HTTExon1Q$_{48}$ΔP2 and the chaperones is disrupted. We have also generated a HTTExon1Q$_{48}$ variant lacking the first proline stretch (HTTExon1Q$_{48}$ΔP1; Fig. 3a). We could previously demonstrate that the first proline stretch of the PRD affects the aggregation propensity of HTTexon1 independently of chaperones. Deletion of the first proline stretch delays the aggregation[8]. To account for that, we performed the FRET fibrilization assay for an expanded time (60 h) to ensure the formation of amyloid fibrils of HTTExon1Q$_{48}$ΔP1 as reflected by the typical sigmoidal fibrilization curve (Fig. 3c, teal curve)[8]. Indeed, we observe a later aggregation onset of HTTExon1Q$_{48}$ΔP1 ($T_{1/2}$ = 26 h; Fig. 3c) compared to HTTExon1Q$_{48}$ ($T_{1/2}$ = 10 h). Notably, although the chaperone complex could not fully suppress the aggregation, it could significantly delay the onset of HTTExon1Q$_{48}$ΔP1 aggregation ($T_{1/2}$ = 35 h; Fig. 3c). As an additional control, we used the ΔN17 mutant (Fig. 3a) of HTTExon1Q$_{48}$ with an intact PRD. Given that the identified binding site (P2 of the PRD) is present in this variant, we predicted that the trimeric chaperone complex should be capable to suppress HTTExon1Q$_{48}$ΔN17 aggregation. Notably, also HTTExon1Q$_{48}$ΔN17 exhibits a slower aggregation kinetic ($T_{1/2}$ = 15 h) than the non-variated HTTExon1Q$_{48}$ (Fig. 3d)[8,11]. To account for that, the assay was also performed for an expanded time (40 h) to ensure the formation of amyloid fibrils. Notably, the trimeric chaperone complex could suppress the fibrilization of HTTExon1Q$_{48}$ΔN17 (Fig. 3d, compare teal and purple curves). These data further support our observation that the PRD and in particular the P2 of HTTExon1Q$_{48}$ represents an additional binding site of chaperones and more specifically for DNAJB1 and Hsc70.

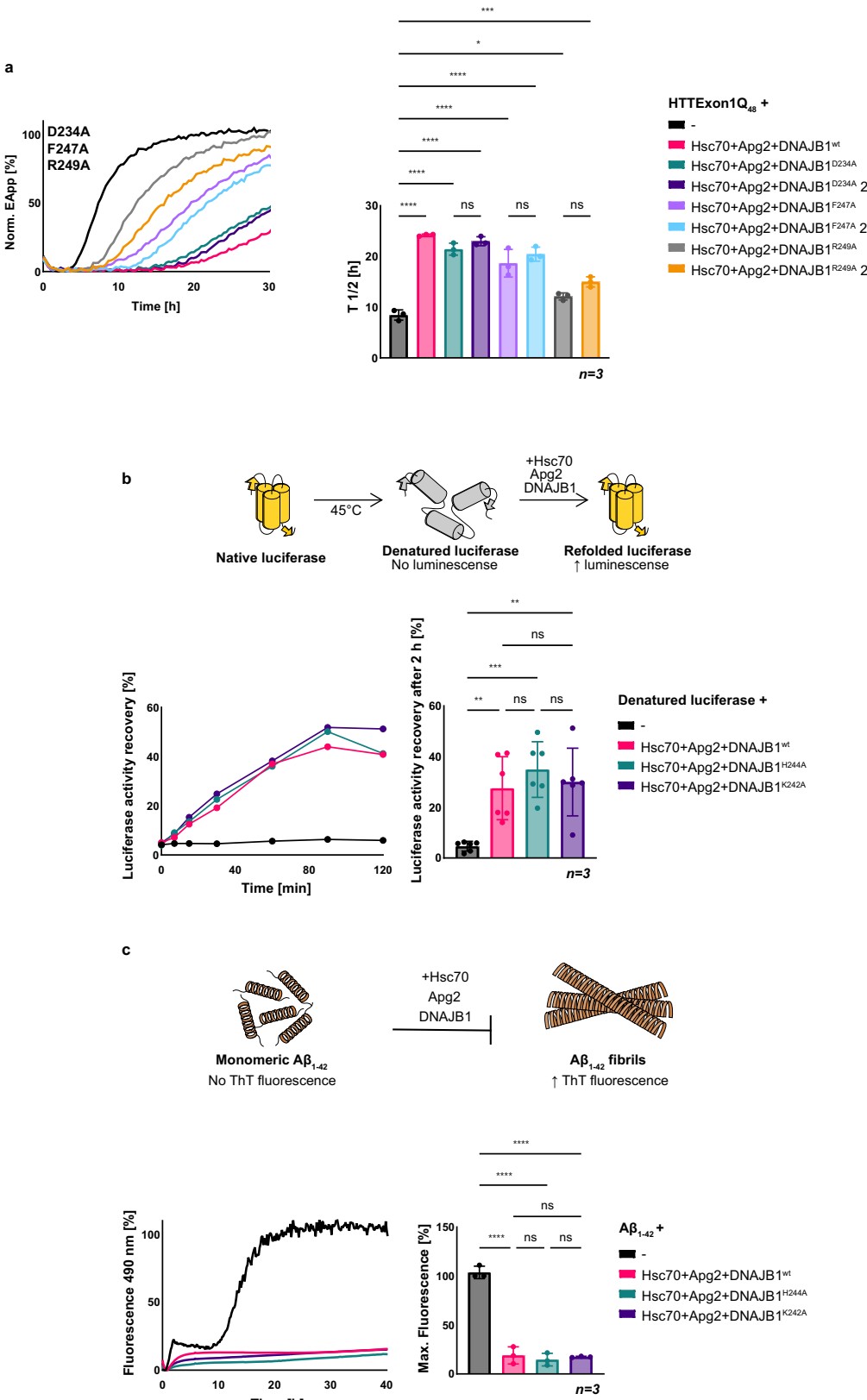

## In silico analysis of the hinge region of DNAJB1

To gain further mechanistic insight into the DNAJB1 structure, its interaction with HTTExon1Q$_{48}$, and the effect of the H244A mutation, we performed in silico analyses using molecular dynamics (MD) simulations. First, a DNAJB1$^{wt}$ model was constructed from a DNAJB1 crystal structure (pdb: 3agz) and simulated in solution for 500 ns. A contact map calculated from this simulation indicates an interaction of the HBM with amino acids of the CTDI and CTDII domain (dark squares in the top half of the contact map in Fig. 4a). A closer look revealed that H244 forms continuous hydrogen bonds to E173 connecting the CTDI with the CTDII (Supplementary Fig. 4a and Fig. 4a, c). The same behavior cannot be observed for DNAJB1$^{H244A}$, since A244 lacks

**Fig. 2 | DNAJB1^[H244A] and DNAJB1^[K242A] do not affect refolding of luciferase or suppression of Aβ$_{1-42}$ aggregation. a** FRET analysis of HTTExon1Q$_{48}$ aggregation and the effect of the addition of Hsc70, Apg2, and DNAJB1^[wt] or variants D234A, F247A, and R249A. Each graph is a representative result of three independent experiments, and a one-way ANOVA analysis of the half-life ($T_{1/2}$) of HTTExon1Q$_{48}$ aggregation under the tested conditions is depicted on the right. Error bars correspond to the SD. ****$P \leq 0.0001$; ***$P \leq 0.001$; *$P \leq 0.1$; ns not significant. **b** Top, schematic representation of a luciferase refolding experiment. Native luciferase is denatured at 45 °C for 15 min and subsequently incubated with chaperones at 20 °C. Refolding of luciferase leads to a recovery of its enzymatic activity that in turn converts luciferin to oxyluciferin which emits luminescence. The increase in luminescence is used as a proxy for refolding efficiency. Bottom, measurement of luminescence emitted over time by heat-denatured luciferase after refolding by Hsc70, Apg2 and DNAJB1^[wt] (magenta), DNAJB1^[H244A] (teal) or DNAJB1^[K242A] (dark purple). The graph is a representative result of three independent experiments and a one-way ANOVA analysis. Bars represent the mean value and error bars correspond to the mean SD. ***$P \leq 0.001$; **$P \leq 0.01$; ns not significant. **c** Top, schematic representation of a Thioflavin T (ThT) assay. Monomeric Aβ$_{1-42}$ forms β-sheet-rich fibrils upon aggregation that are bound by ThT leading to an increased emission of fluorescence. Suppression of Aβ$_{1-42}$ fibrilization was assessed by adding Hsc70, Apg2, and DNAJB1^[wt] (magenta), DNAJB1^[H244A] (teal) or DNAJB1^[K242A] (dark purple) to monomeric Aβ$_{1-42}$. A graph depicting a representative result of three independent experiments is shown on the bottom left, and the corresponding one-way ANOVA analysis of the final fluorescence intensity of ThT under the tested conditions of three independent experiments is depicted on the right. Bars represent the mean value, and error bars correspond to the mean SD. ****$P \leq 0.0001$; ns not significant.

hydrogen bonding donors that could reach E173, resulting in reduced contacts between the HBM (especially A244) and the CTDI domain (residues 170–175) (differential contact map on the right in Fig. 4a, where positive values indicate increased interactions and negative values decreased interactions). Notably, mutation of E173 does not affect the activity of DNAJB1 in suppressing HTTExon1Q$_{48}$ fibrilization together with Hsc70 and Apg2 (Fig. 4b), whereas mutation of the adjacent E174 by alanine abrogates the capacity of DNAJB1 to suppress HTTExon1Q$_{48}$ aggregation similarly as for DNAJB1^[H244A] together with Hsc70 and Apg2 (Fig. 4b). Both glutamate residues (E173 and E174) are highly conserved among class A and B J-domain proteins containing a HBM (Fig. 4b). However, E174 is not located in the HTTExon1Q$_{48}$ binding region, but forms a stable hydrogen bond network with S171, C179, and K181 (Supplementary Fig. 4b), which have been shown to bind to Hsc70 in NMR experiments[23]. We, therefore, postulate that mutation of E174 by alanine perturbs this H-bond network and affects the remodeling of HTT by DNAJB1/Hsc70. In a similar way, substitutions of H244 by phenylalanine or alanine could destabilize the HTTExon1Q$_{48}$/DNAJB1 complex. This is investigated in further MD simulations as described below.

## Computational exploration of minimum free energy structures of HTTExon1Q$_{48}$

To structurally analyze the complex formed between DNAJB1^[wt] or its mutation variants and HTTExon1Q$_{48}$, we first needed to predict, refine and then dock a HTTExon1Q$_{48}$ model to DNAJB1. There is little structural information available for HTTExon1Q$_{48}$. An initial starting structure was predicted by the homology modeling algorithm I-TASSER[24,25], serving as input structure followed by more intensive refinement via the enhanced-sampling TIGER2h algorithm[26]. The enhanced-sampling simulations explore the peptide phase space independently of the input structures. Five distinct structural clusters for HTTExon1Q$_{48}$ were extracted via dihedral principal component analysis (dPCA)[27]. Two clusters compete for the minimum energy structure (black and purple regions of the PCA map in Supplementary Fig. 3a). All identified clusters share a common structural feature, in that the polyQ domain (depicted in orange, Supplemental Fig. 4b) is mainly helical with one long or multiple short helices. The remaining C-terminal fragment (PRD with P1 (aa 65–74) and P2 (aa 93–102) depicted in red; the sequence between P1 and P2 is depicted in black) coordinates to that helix and is slightly restrained through intramolecular interactions. The calculated root mean square fluctuation (RMSF) in all clusters is high, due to the very flexible PRD following the polyQ domain. Adequate phase-space sampling was ensured by the convergence of the cluster probability distributions and of the secondary structure content in the five most important clusters (Supplementary Fig. 3c).

## In silico study of the DNAJB1/HTTExon1Q$_{48}$ complex

All five identified clusters of HTTExon1Q$_{48}$ were docked to DNAJB1^[wt], employing the rigid-docking server HDOCK. Only clusters 1, 2, and 5 lead to a reasonable complex formation with DNAJB1^[wt], involving the HBM and P2 in the binding interface (Supplementary Fig. 3c). The P2 domain in clusters 3 and 4 is partially shielded by the N17 and polyQ domain (Supplementary Fig. 3b). The three assemblies (clusters 1, 2, and 5 with DNAJB1^[wt]) were further subjected to 500 ns of MD simulation. Only clusters 1 and 2 were observed to form a stable complex with DNAJB1^[wt] (see WT model and contact map in Fig. 4c), whereas cluster 5 quickly detaches from HTTExon1Q$_{48}$ (Supplementary Fig. 3c). Major contacts of amino acids of the P2 of the PRD were formed to P243, H244, N245, I246, and K248 of DNAJB1^[wt]. Interestingly, due to hydrogen bonding between E173 and the H244 side chain, only the backbone atoms of H244 are facing P2 of HTTExon1Q$_{48}$ (see enlarged WT molecular model in Fig. 4c).

We then docked clusters 1 and 2 of HTTExon1Q$_{48}$ to the three DNAJB1 variants: DNAJB1^[H244A], DNAJB1^[E173A], and DNAJB1^[E174A]. Interestingly, the clusters soon detach and move away from the HBM of DNAJB1^[H244A], leading to a very faint contact map (Fig. 4c, right-hand side). Due to the missing hydrogen bond between residues 173 and 244, the alanine side chain at position 244 rotates outwards, preventing its backbone to firmly bind to HTTExon1Q$_{48}$.

Instead, similar contact maps as for DNAJB1^[wt] are observed for the DNAJB1^[E173A] and DNAJB1^[E174A] variants (Fig. 4c), with D241, P243, H244, and N245 staying in close contact to the P2 of the PRD. In the HTTExon1Q$_{48}$/DNAJB1^[E173A] complex, no hydrogen bond forms between A173 and H244, so that the histidine side chain rotates outwards. However, now it is the side chain of H244 that forms stable hydrogen bonds to the backbone atoms of P2. In the simulation of HTTExon1Q$_{48}$/DNAJB1^[E174A], H244 does not form a hydrogen bond to E173 (which binds instead to N245, see Fig. 4c), but with the backbone atoms of prolines in the P2 of HTTExon1Q$_{48}$ (see detailed interactions in the molecular models in Fig. 4c).

In summary, our in silico analysis fully supports the experimental finding that the P2 region of HTTExon1Q$_{48}$ forms stable contacts with the HBM of DNAJB1, including residues 238–246 (Fig. 1a). H244 plays a fundamental role in stabilizing the complex via hydrogen bond interactions of either its backbone (as observed in the WT model) or its side chain (as observed e.g. in the E173A and E174A variants) to HTTExon1Q$_{48}$. Substitutions of H244 by alanine break these bonds and destabilize the complex, explaining the inability of DNAJB1^[H244A] to suppress HTT fibrilization together with Hsc70 and Apg2.

## One protomer of the DNAJB1 dimer is sufficient for suppressing HTTExon1Q$_{48}$ fibrilization with Hsc70 and Apg2

The MD simulations of the interaction of HTTExon1Q$_{48}$ with the different DNAJB1 dimers suggest that each protomer could in theory interact with a HTTExon1Q$_{48}$ protein as the bound HTTExon1Q$_{48}$ protein faces outward of the arch formed by the DNAJB1 dimer (Fig. 4c and Supplementary Fig. 3d). Is the identified binding site (HBM) of both DNAJB1 protomers required for the suppression of HTTExon1Q$_{48}$ fibrilization? Does DNAJB1 form a dimer in its active state as chaperone? To address these questions, we employed a chemically induced dimerization approach that allows us to generate DNAJB1 dimers that

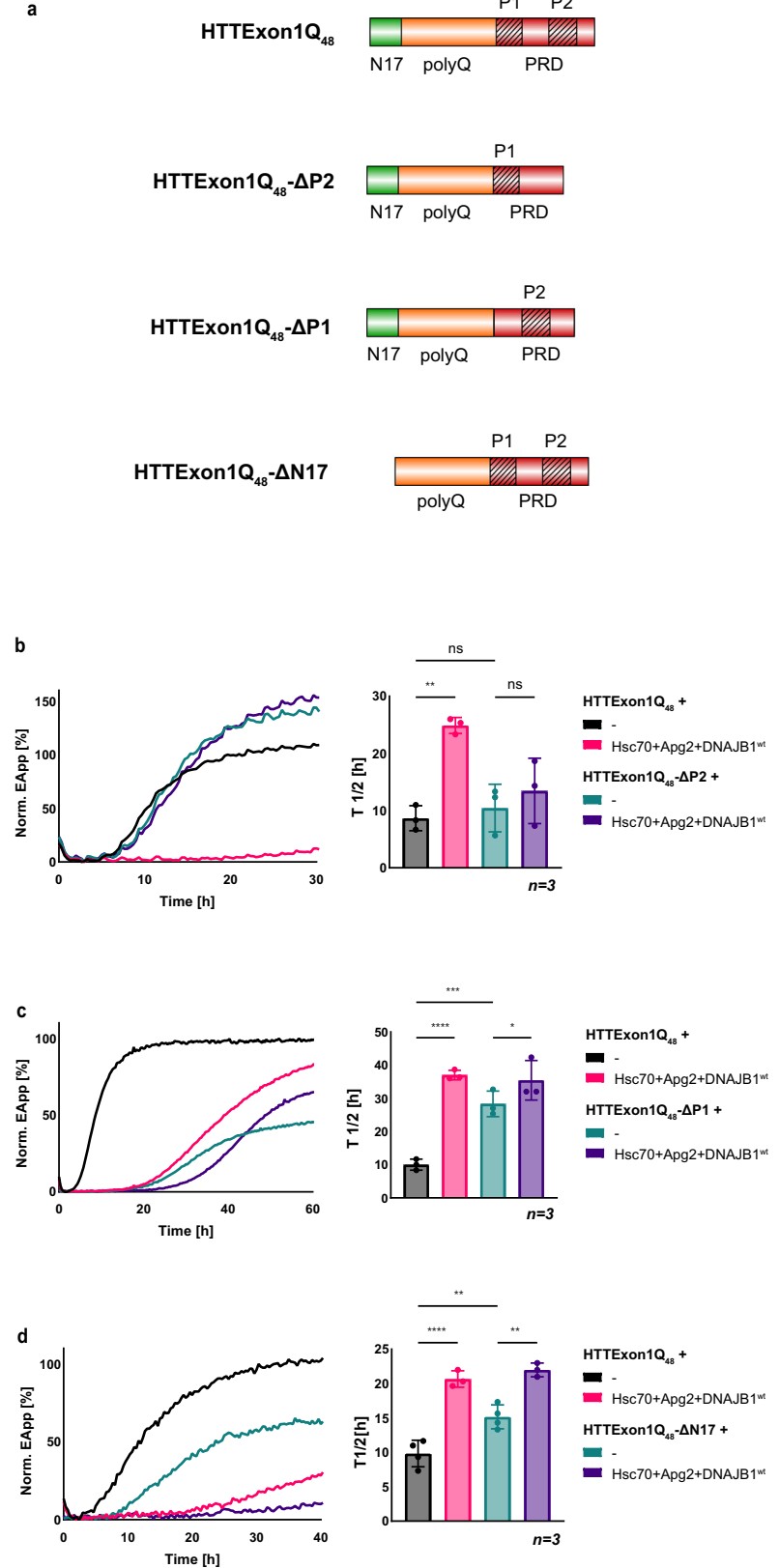

**Fig. 3 | DNAJB1 interacts with the P2 region within the PRD of HTTExon1Q48.**
**a** Schematic representation of HTTExon1Q48 with its N17, polyQ, and PRD domains and the three variants, HTTExon1Q48-ΔP2 that lacks the P2 region, HTTExon1Q48-ΔP1 lacking the P1 region, and HTTExon1Q48-ΔN17 with a missing N17 domain.
**b–d** FRET assay of the aggregation of (**b**) HTTExon1Q48-ΔP2, (**c**) HTTExon1Q48-ΔP1, and (**d**) HTTExon1Q48-ΔN17 compared to HTTExon1Q48 and upon addition of Hsc70, Apg2, and DNAJAB1wt. The graphs are representative results of three independent experiments and one-way ANOVA analyses of the half-life ($T_{1/2}$) of HTTExon1Q48-ΔP2, HTTExon1Q48-ΔP1, and HTTExon1Q48-ΔN17 compared to HTTExon1Q48 aggregation under the tested conditions are shown on the right. Bars represent the mean value, and error bars correspond to the mean SD.
$****P \leq 0.0001$; $***P \leq 0.001$; $**P \leq 0.01$; $*P \leq 0.1$; ns not significant.

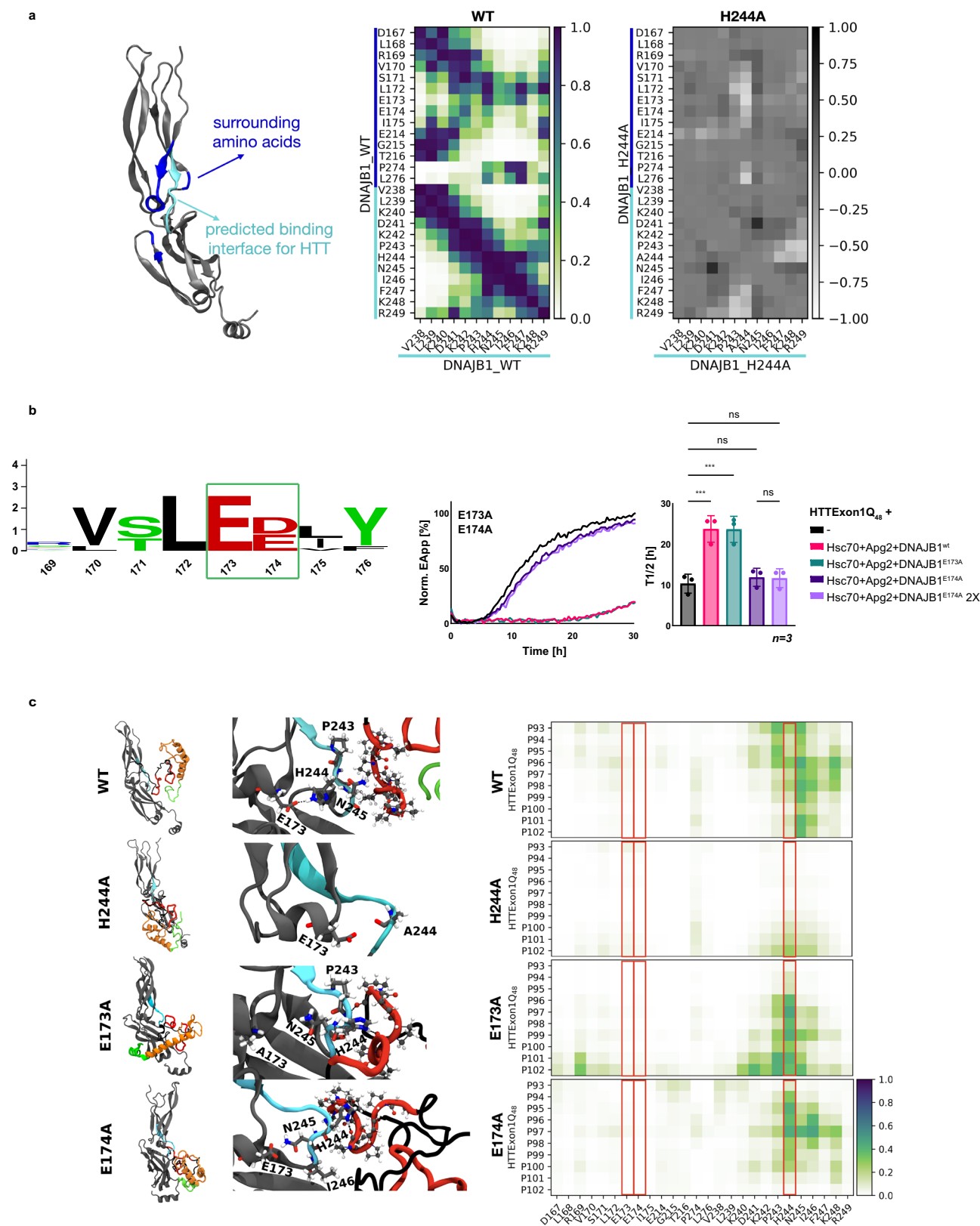

contain the H244A mutation in only one or in both protomers. For that, we fused DNAJB1 lacking the dimerization domain (DNAJB1$^{\Delta DD}$) with either FK506 binding protein (FKBP12) or the FKBP-rapamycin binding domain of mTOR (FRB). FKBP12 and FRB interact in the presence of Rapamycin and form a chemically-inducible dimer (CID) (Fig. 5a)[28]. We first confirmed the dimerization of the DNAJB1$^{\Delta DD}$-FKBP12 and

DNAJB1$^{\Delta DD}$-FRB in the presence of Rapamycin using size-exclusion chromatography (Supplementary Fig. 2d). We then formed dimers of DNAJB1$^{wt}$ (DNAJB1-CID$^{wt/wt}$), DNAJB1$^{H244A}$ on both protomers (DNAJB1-CID$^{H244A/H244A}$) and mixed DNAJB1 dimers with only one mutated protomer (DNAJB1-CID$^{wt/H244A}$ or DNAJB1-CID$^{H244A/wt}$). The generated CID of DNAJB1 mutants are schematically depicted in Fig. 5b. Next, we tested

**Fig. 4 | Molecular dynamic simulation of DNAJB1 variants, interacting with HTTExon1Q$_{48}$. a** Atomistic structure of monomeric DNAJB1$^{wt}$. Contact maps of DNAJB1$^{wt}$ and DNAJB1$^{H244A}$, representing contacts of the HBM with surrounding amino acids. Positive values indicate increased and negative decreased interactions within DNAJB1$^{H244A}$ compared to DNAJB1$^{wt}$. **b** Left, conservation logo of the aligned sequences of class A and B JDP containing a HBM reveal highly conserved E173 and E174 (green square). Right, HTTExon1Q$_{48}$ aggregation analysis upon addition of Hsc70, Apg2, and DNAJB1$^{wt}$ or DNAJB1$^{E173A}$ and DNAJB1$^{E174A}$. Each graph is a representative result of three independent experiments. A one-way ANOVA analysis of the $T_{1/2}$ of HTTExon1Q$_{48}$ aggregation is depicted on the right. Bars represent the mean value and error bars correspond to the mean SD. ***$P \leq 0.001$; ns not significant. **c** MD snapshots of monomeric DNAJB1$^{wt}$ or DNAJB1$^{H244A}$, DNAJB1$^{E173A}$, and DNAJB1$^{E174A}$ in complex with HTTExon1Q$_{48}$ (left). DNAJB1$^{wt}$, DNAJB1$^{H244A}$, and DNAJB1$^{E174A}$ with cluster 1 and DNAJB1$^{E173A}$ with cluster 2 of HTTExon1Q$_{48}$. Only one monomer was visualized for simplicity, but docking and simulations were performed with dimeric DNAJB1. Atomistic structure of the HBM of DNAJB1$^{wt}$ or variants and P2 of HTTExon1Q$_{48}$, interacting via hydrogen bonds (dotted lines) or non-bonding interactions (middle). DNAJB1 (gray) with highlighted amino acids with a coloring code according to the atom types: hydrogen (white), carbon (cyan), oxygen (red), nitrogen (blue). The HBM is colored in cyan. The domains of HTTExon1Q$_{48}$ are indicated by different colors: polyQ (orange), PRD (P1 and P2: red; residues between P1 and P2: black), N17 (green) and highlighted amino acids in CPK style. Contact maps of DNAJB1$^{wt}$ or variants and HTTExon1Q$_{48}$, focus on the HBM and P2 domain (left). Contact maps were constructed by averaging the contacts over the last 400 ns of the MD simulation and additionally averaging over interactions recorded for cluster 1 and 2 of HTTExon1Q$_{48}$ with DNAJB1. Distances were converted by a rational switching function, defining the contact distance (inflection point) at 1 nm, where a value of 1 represents a close contact and 0 no contact. Residues 173, 174, and 244 are highlighted with a red box.

all DNAJB1-CID variants in the FRET HTTExon1Q$_{48}$ fibrilization assay and first confirmed that DNAJB1-CID$^{wt/wt}$ was as effective in suppressing HTTExon1Q$_{48}$ fibrilization as DNAJB1$^{wt}$ together with Hsc70 and Apg2 (Fig. 5c, top, compare magenta and teal curves). We then tested the double and single H244A mutants and observed that the double mutant (H244A in both protomers) was as expected most affected and shows the least suppression of HTTExon1Q$_{48}$ fibrilization together with Hsc70 and Apg2 (dark-purple curve). The single mutations that only harbor the H244A mutation in one protomer show an intermediate phenotype. They are both less effective than DNAJB1$^{wt/wt}$, but significantly more active in suppressing HTTExon1Q$_{48}$ with Hsc70 and Apg2 than DNAJB1-CID$^{H244A/H244A}$ (Fig. 5c, top, compare light purple and light-blue curves with the dark-purple curve). From these data, we conclude that one active protomer of the DNAJB1 dimer is sufficient to suppress HTTExon1Q$_{48}$ aggregation with Hsc70 and Apg2. However, doubling the concentration of DNAJB1-CID$^{wt/H244A}$ or DNAJB1-CID$^{H244A/wt}$ to provide the same number of intact protomers as in DNAJB1-CID$^{wt/wt}$ did not lead to a further increase of suppression of HTTExon1Q$_{48}$ aggregation (Fig. 5c, bottom).

### Monomeric DNAJB1 is more potent than dimeric DNAJB1 in HTTExon1Q$_{48}$ suppression with Hsc70 and Apg2

DNAJB1 is purified as a stable dimer (Fig. 6a). However, it is not known if the dimeric state of DNAJB1 is its active functional state or if the dimer constitutes a storage form that disassembles upon binding to substrates or partner chaperones. A subgroup of class B JDPs, including DNAJB6, contains a short CTD with a S/T-rich domain that mediates the formation of oligomers. These oligomers serve as a storage form and upon activation, DNAJB6 monomers dynamically associate and dissociate from the oligomers[29]. We wondered if a similar behavior could be observed for DNAJB1. We hence deleted the dimerization domain of DNAJB1 and first confirmed that DNAJB1$^{\Delta DD}$ is monomeric by size-exclusion chromatography (Fig. 6a). Next, we tested DNAJB1$^{\Delta DD}$ in its capacity to suppress HTTExon1Q$_{48}$ fibrilization together with Hsc70 and Apg2. Interestingly, DNAJB1$^{\Delta DD}$ is significantly more potent in suppressing HTTExon1Q$_{48}$ fibrilization with Hsc70 and Apg2 than DNAJB1$^{wt}$ that forms a dimer (Fig. 6b). The $T_{1/2}$ of HTTExon1Q$_{48}$ fibrilization was enhanced to 37 h in the presence of Hsc70, Apg2 + DNAJB1$^{\Delta DD}$ vs 26 h in the presence of DNAJB1$^{wt}$. Notably, the lack of the dimerization domain of DNAJB1 had no significant effect on the refolding of denatured firefly luciferase (Fig. 6c). We assessed DNAJB1$^{\Delta DD}$ also in the suppression of A$\beta_{1-42}$ aggregation, however as A$\beta_{1-42}$ aggregation is already fully suppressed by Hsc70, Apg2 together with DNAJB1$^{wt}$, an additional positive effect of the $\Delta$DD variant could not be assessed (Fig. 6d). We conclude that monomeric DNAJB1 is more potent in suppressing HTTExon1Q$_{48}$ fibrilization with Hsc70 and Apg2 and that this higher activity as monomer appears to be substrate-specific.

### The G/F-rich region of DNAJB1 contributes to the suppression of HTTExon1Q$_{48}$ aggregation with Hsc70 and Apg2

The identified HBM is to a certain degree conserved in JDPs (Fig. 1b), including the class A J-domain protein DNAJA1. The histidine and lysine residues are located in the same position as in DNAJB1 (Fig. 7a). Yet, DNAJA1 cannot suppress HTTExon1Q$_{48}$ fibrilization together with Hsc70 and Apg2 (Fig. 7b, compare teal and magenta curves). This observation implies that additional factors that are present in DNAJB1 but not in DNAJA1 and/or factors that are present in DNAJA1 but not in DNAJB1, contribute to or interfere with the chaperone activity towards HTTExon1Q$_{48}$. The G/F-rich domain of class A JDPs is shorter than the one of class B. In addition, class A JDPs contain a Zn-finger-like region (ZFLR) that protrudes from the CTDI, while class B JDPs do not harbor ZFLRs (Supplementary Fig. 5a). To test the contribution of both elements, we engineered a DNAJB1ized version of DNAJA1 (DNAJA1$^{JB1ized}$) by deleting its ZFLR and replacing its G/F-rich domain with that of DNAJB1 (Fig. 7a). We first checked the integrity of the chimeric protein by analyzing its ability to induce the ATPase activity of Hsc70 and to support the refolding of heat-denatured luciferase together with Hsc70 and Apg2. We did not observe any significant defects in these control assays (Supplementary Figs. 2b and 5b). We then tested the chimeric protein in the FRET HTTExon1Q$_{48}$ fibrilization assay. Notably, DNAJA1$^{JB1ized}$ could delay the aggregation of HTTExon1Q$_{48}$ together with Hsc70 and Apg2 by 10 h (Fig. 7b, compare purple and teal curves). Interestingly, deletion of the ZFLR of DNAJA1 results in a CTD that resembles that of DNAJB1. However, DNAJA1$^{\Delta ZFLR}$ is still inactive towards HTTExon1Q$_{48}$ as well as luciferase (Supplementary Fig. 5c). These data suggest that the G/F-rich region of DNAJB1 could be a critical domain for the suppression of HTTExon1Q$_{48}$ aggregation together with Hsc70 and Apg2. Indeed, deletion of the G/F-rich region of DNAJB1 or substitution with the G/F-rich region of DNAJA1 completely abrogated the activity of DNAJB1 in the suppression of HTTExon1Q$_{48}$ fibrilization and also in the refolding of heat-denatured luciferase together with Hsc70 and Apg2 (Supplementary Fig. 5c). Nevertheless, based on the data obtained using the chimeric DNAJA1$^{JB1ized}$ we can conclude that apart from the HBM, the G/F-rich region of DNAJB1 contributes to the suppression of HTTExon1Q$_{48}$ aggregation together with Hsc70 and Apg2.

### DNAJB1$^{H244A}$ is severely compromised in the binding and disaggregation of HTTExon1Q$_{48}$ fibrils

We could previously establish that HTTExon1Q$_{48}$ fibrils can be disaggregated by the same trimeric chaperone complex Hsc70, DNAJB1, and Apg2 in an ATP-dependent manner[18]. Structural data and models of amyloid fibrils formed by HTTExon1Q$_{46}$ suggest that the PRD forms highly flexible structures that protrude outwards from the fibrillar core and could hence be accessible by the chaperones[30]. It is therefore possible that the same binding interface is used for soluble as well as

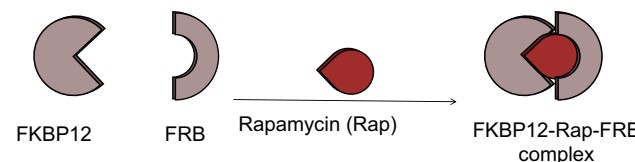

**a**  FKBP12-FRB chemically-inducible dimerization (CID)

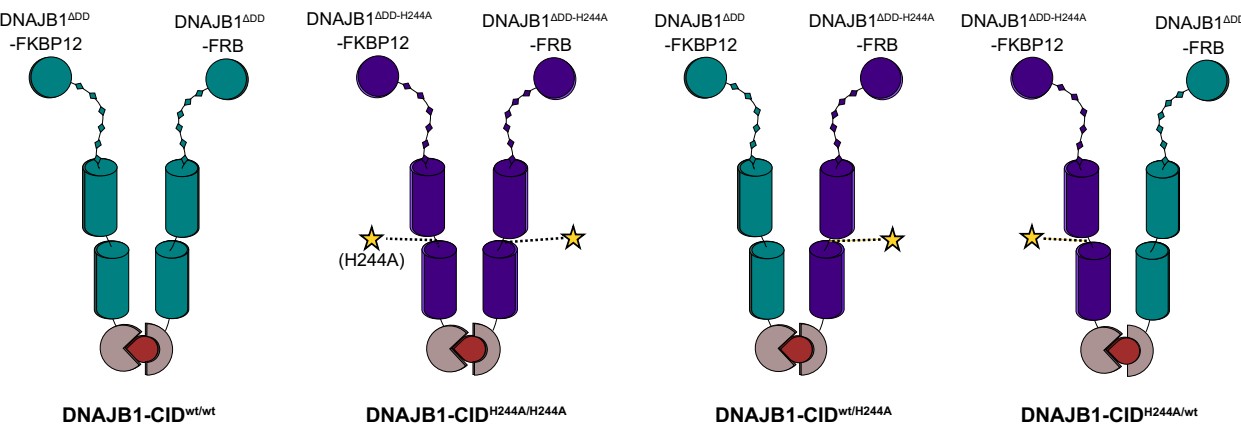

**b**  DNAJB1 chemically-inducible dimers

**c**

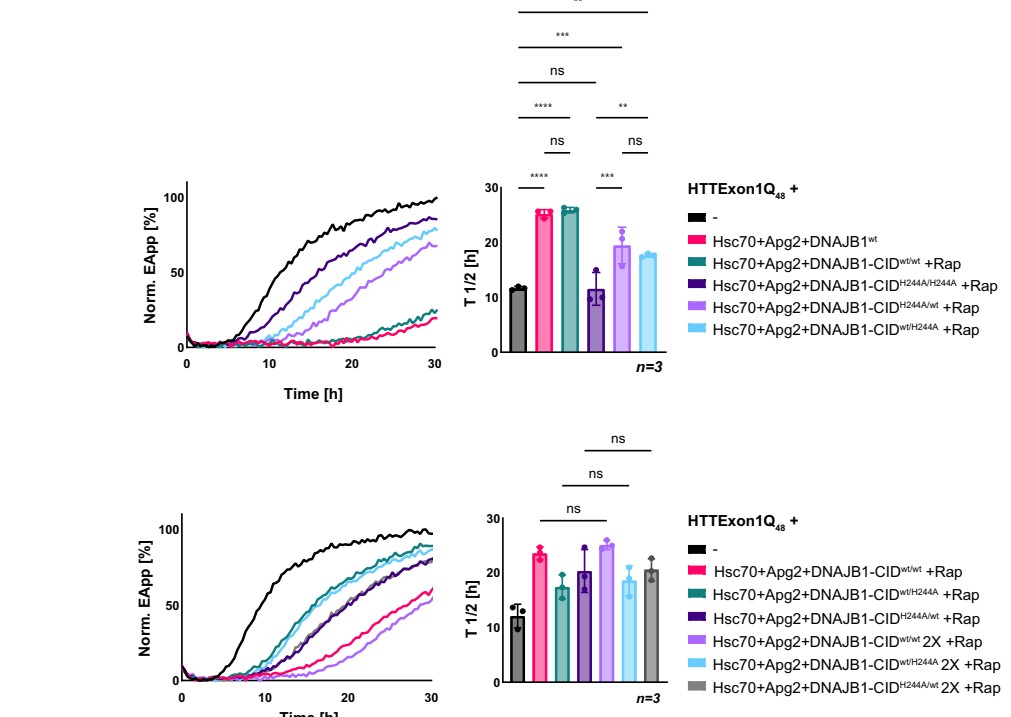

fibrillar HTT by DNAJB1. Indeed, amyloid fibrils of HTTExon1Q$_{48}$ΔP2 that lack the P2 region of the PRD cannot be disaggregated by the three chaperones (Fig. 8a, c).

Next, we tested the ability of DNAJB1$^{H244A}$ to bind to HTTExon1Q$_{48}$ fibrils. The affinity of DNAJB1 to HTT fibrils is much higher than to the soluble form of HTT, similar to previous results for α-synuclein[31]. The

stronger interaction with HTT fibrils enables us to compare binding affinities between DNAJB1$^{wt}$ and DNAJB1$^{H244A}$. Notably, we observed a severe reduction of HTTExon1Q$_{48}$ fibril binding to about 30% compared to DNAJB1$^{wt}$ (Fig. 8b). Taken together, these data support the assumption that the P2 region is also the binding site for DNAJB1 in HTT fibrils and that DNAJB1 uses a similar binding mode for soluble as

**Fig. 5 | Protomers of DNAJB1 act independently in the suppression of HTTExon1Q$_{48}$ aggregation with Hsc70 and Apg2. a** Schematic representation of a chemically induced dimerization (CID) system. FKBP12 and FRB heterodimerize upon the addition of Rapamycin. **b** Schematic representation of DNAJB1 chemically induced dimers. DNAJB1$^{\Delta DD}$ (teal) and DNAJB1$^{\Delta DD\text{-}H244A}$ (dark purple) were expressed as fusion constructs with either FKBP12 or FRB. The addition of Rapamycin facilitates dimer formation of DNAJB1-CID$^{wt/wt}$ (teal), DNAJB1-CID$^{H244A/H244A}$ (dark purple), DNAJB1-CID$^{wt/H244A}$ (mixed protomers) and DNAJB1-CID$^{H244A/wt}$ (mixed protomers). **c** Top, FRET analysis of HTTExon1Q$_{48}$ aggregation upon addition of Hsc70, Apg2, and DNAJB1$^{wt}$ or DNAJB1-CIDs in the presence of Rapamycin (Rap): DNAJB1-CID$^{wt/wt}$ (teal), DNAJB1-CID$^{H244A/H244A}$ (dark purple), DNAJB1-CID$^{wt/H244A}$ (light blue), DNAJB1-CID$^{H244A/wt}$ (light purple). Bottom, FRET analysis of HTTExon1Q$_{48}$ aggregation upon addition of single or double the concentration (2x) of DNAJB1-CIDs in the presence of Rap: DNAJB1-CID$^{wt/wt}$ (magenta and light purple), DNAJB1-CID$^{wt/H244A}$ (teal and light blue), DNAJB1-CID$^{H244A/wt}$ (dark purple and gray).The graphs are a representative result of three independent experiments and a one-way ANOVA analysis of the half-life ($T_{1/2}$) of HTTExon1Q$_{48}$ aggregation under the tested conditions is depicted on the right. Bars represent the mean value and error bars correspond to the mean SD. ****$P \leq 0.0001$; ***$P \leq 0.001$; **$P \leq 0.01$; ns not significant.

well as aggregated HTTExon1Q$_{48}$. We hence hypothesized that DNAJB1$^{H244A}$ is also compromised in the disaggregation of HTTExon1Q$_{48}$ amyloid fibrils together with Hsc70 and Apg2. The addition of DNAJB1$^{H244A}$ and Hsc70 + Apg2 to preformed HTTExon1Q$_{48}$ fibrils led to a re-solubilization of HTTExon1Q$_{48}$ of only approximately 50% compared to DNAJB1$^{wt}$ + Hsc70 and Apg2 (Fig. 8d). The effect of the DNAJB1$^{H244A}$ was less pronounced in the disaggregation assay compared to the suppression of aggregation assay where a single point mutation of H244 (DNAJB1$^{H244A/F}$) led to a complete abrogation of the chaperone activity towards soluble HTTExon1Q$_{48}$ (Fig. 1d, e). We tested an additional substitution of H244 with phenylalanine that represents an equally hydrophobic but more bulky amino acid in the disaggregation assay. Notably, DNAJB1$^{H244F}$ together with Hsc70 and Apg2 did not exhibit any disaggregation activity towards HTTExon1Q$_{48}$ fibrils (Fig. 8d). DNAJB1$^{H244F}$ was, however, as active as DNAJB1$^{wt}$ in luciferase refolding, further demonstrating the specificity of H244 for HTT and integrity of the mutant protein (Supplementary Fig. 2c). Analogous to our observations on the suppression of HTTExon1Q$_{48}$ fibrilization, DNAJB1$^{K242A}$ was less impaired and led to a re-solubilization of 70% of HTTExon1Q$_{48}$ fibrils compared to DNAJB1$^{wt}$ together with Hsc70 and Apg2 (Fig. 8d). These data demonstrate that H244 and to a lesser extent K242 of DNAJB1 are important for the disaggregation of HTTExon1Q$_{48}$ amyloid fibrils.

Interestingly, the dimerization mutant of DNAJB1 that showed a higher activity together with Hsc70 and Apg2 in the suppression of HTTExon1Q$_{48}$ fibrilization (Fig. 6b) did not show any benefit in the disaggregation of HTTExon1Q$_{48}$ fibrils (Fig. 8d). Analogous to the suppression of HTTExon1Q$_{48}$ fibrilization, DNAJA1 showed no activity toward HTTExon1Q$_{48}$ fibrils and DNAJA1$^{JBlized}$ exhibited only a slight, yet not significant increase in resolubilized HTTExon1Q$_{48}$ together with Hsc70 and Apg2 (Fig. 8e, f). In summary, these data suggest that DNAJB1 uses the P2 region of the PRD as the same epitope within HTT for HTTExon1Q$_{48}$ suppression of fibrilization and the disaggregation of fibrils. In addition, the HBM and in particular H244 of DNAJB1 is a key residue for both, suppression as well as disaggregation of HTT fibrils. However, our data on the role of the dimerization domain and the G/F-rich region of DNAJB1 suggest that there are also mechanistic differences in the remodeling of soluble and aggregated HTT.

**Overexpression of DNAJB1$^{wt}$, but not DNAJB1$^{H244A}$ rescues HTTExon1Q$_{97}$-GFP aggregation in cell culture**

To assess the biological significance of the identified HBM within DNAJB1, we expressed HTTExon1Q$_{97}$-GFP in HEK293 cells. HTTExon1Q$_{97}$-GFP rapidly aggregates and forms visible foci (Fig. 9c) that can also be detected by filter-retardation analysis (Fig. 9b). Overexpression of DNAJB1$^{wt}$ (3-fold over endogenous DNAJB1 levels) rescued the aggregation of HTTExon1Q$_{97}$ as shown previously by us[18]. Importantly, co-expression of DNAJB1$^{H244A}$ did not show any rescue of HTTExon1Q$_{97}$-GFP aggregation as demonstrated by fluorescence imaging and filter retardation (Fig. 9a–c). The expression levels of DNAJB1$^{wt}$ and DNAJB1$^{H244A}$ were comparable and did not affect the expression of HTTExon1Q$_{97}$-GFP either (Fig. 9a). Depletion of DNAJB1 expression using siRNA led, as expected, to an increase in HTTExon1Q$_{97}$ aggregation that could be rescued by overexpression of DNAJB1$^{wt}$ but not of

DNAJB1$^{H244A}$ (Fig. 9d, e). These data demonstrate that a mutation of a single amino acid of DNAJB1 can completely abrogate its chaperone activity towards HTTExon1Q$_{97}$-GFP in cultured cells.

## Discussion

In this study, we have identified and characterized an interaction interface between DNAJB1/Hsc70 and HTTExon1. We could map the binding site within DNAJB1 to the hinge region between CTDI and CTDII. The highly conserved H244 plays a vital role. Mutation of this single amino acid completely abrogates the suppression of aggregation and disaggregation of HTT by the trimeric chaperone complex, Hsc70 + Apg2 + DNAJB1. Importantly, mutation of DNAJB1$^{H244}$ did not affect the chaperone activity towards other protein substrates, demonstrating its specificity for HTT.

Within HTTExon1Q$_{48}$, we could map the binding of DNAJB1 to the PRD and more specifically and predominantly to the second proline stretch (P2) of the PRD. Interestingly, Hsc70 was found to bind to the very same site within HTTExon1 as DNAJB1. As we could only detect an interaction of Hsc70 with the PRD when DNAJB1 was present, it is tempting to speculate that the binding of both chaperones to the P2 site is mutually exclusive and may play a role in the HTT transfer from DNAJB1 to Hsc70 within the chaperone cycle. Notably, the HBM within DNAJB1 does not overlap with the binding interface to Hsc70 (Fig. 7c, left) and it is thus feasible that DNAJB1, Hsc70, and HTT form a transient ternary complex for the transfer of HTT from DNAJB1 to Hsc70. Analogously, the HTT-binding site within Hsc70 does not overlap with the Apg2 binding site (Fig. 7c, right). Thus, Apg2 could bind to the Hsc70-HTT complex to facilitate the exchange of ADP by ATP that in turn leads to the dissociation of HTT when Hsc70 adopts the open conformation (see below).

With the identification of the PRD, we add a binding site for chaperones to the previously identified interaction sites on HTTExon1 that cluster to the N17 and to the polyQ domains[11,12,14,15]. The PRD has been shown to be protective against the aggregation propensity of the polyQ domain. Recent studies, however, demonstrated the importance of a fine-tuned balance between the polyQ expansion and the poly-proline domain[8]. A concomitant expansion of the first proline stretch (P1) counteracts the aggregation and toxicity of HTTExon1 with an expanded polyQ stretch. The contribution of P2 remained so far enigmatic. We could now demonstrate that the P2 stretch of the PRD represents the binding site for DNAJB1 to alleviate the aggregation of HTT together with Hsc70 and Apg2.

Our in silico analysis of HTTExon1Q$_{48}$/DNAJB1 complexes sheds light into atomistic details of the binding interface. In particular, a histidine at position 244 is crucial to enable the formation of stable complexes via H-bonds either through its backbone or its side chain. This can explain why DNAJB1$^{H244R}$ and DNAJB1$^{H244Q}$, which can undergo H-bond interactions, still present some ability to prevent HTT fibrilization with Hsc70 and Apg2, whereas DNAJB1$^{H244A}$ and DNAJB1$^{H244F}$ do not, due to the hydrophobic character of the side chain at position 244. Further studies are required to elucidate the precise function of E174, although our simulations suggest that it is not involved in the binding of HTTExon1Q$_{48}$ but might rather play a role in the interplay of HTTExon1Q$_{48}$ with DNAJB1 and Hsc70. As

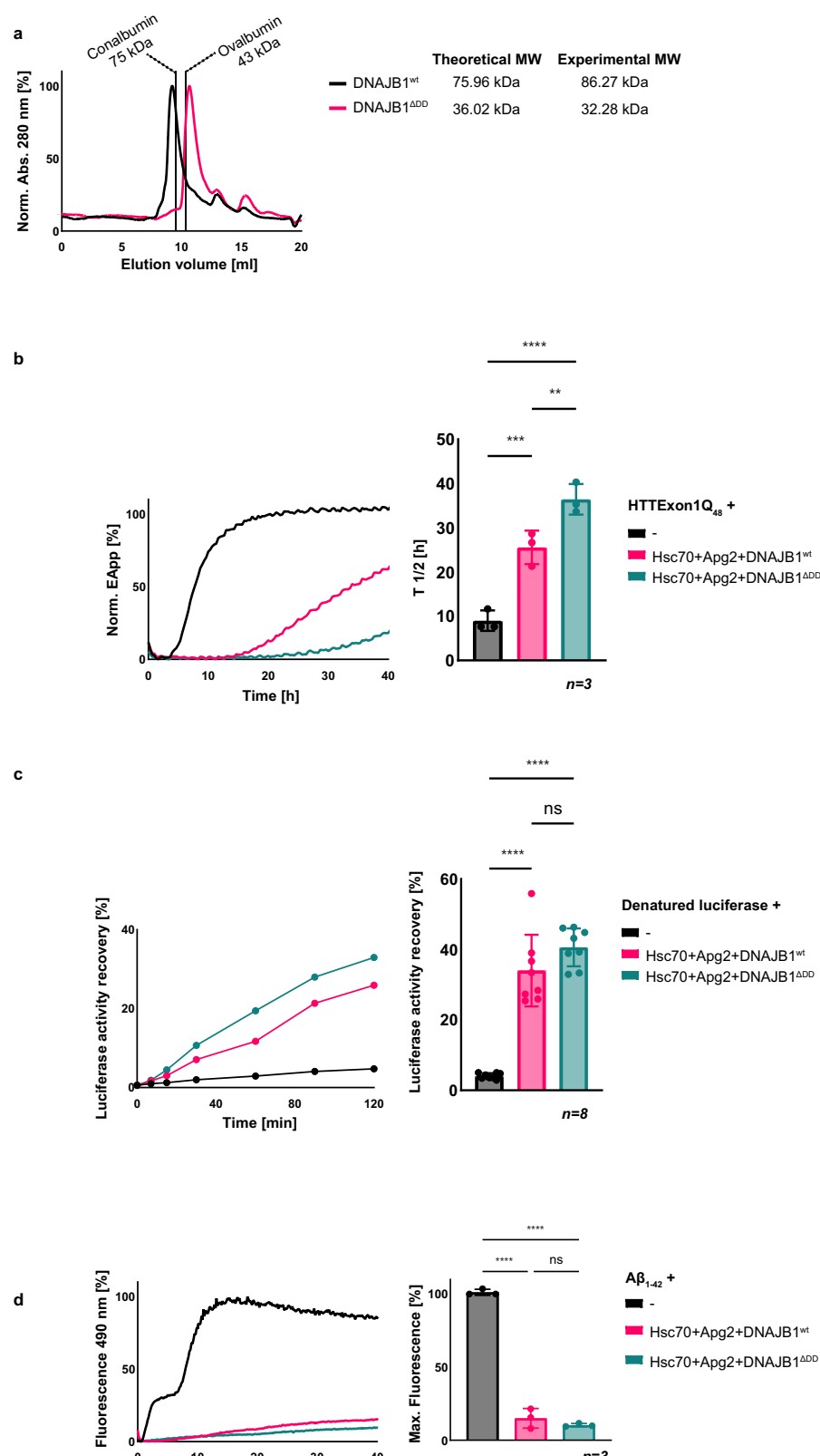

DNAJB1[E174A] does not show defects in its ability to induce the ATPase activity of Hsc70 and can refold luciferase with Hsc70 and Apg2, E174 specifically affects HTT and might be required for the transfer of HTTExon1Q$_{48}$ from DNAJB1 to Hsc70.

Despite the rational explanation provided by the simulations, we acknowledge also their limitations. In particular, they do not include explicitly Hsc70, Apg2, and ATP, roots upon a de novo structural prediction for the HTTExon1Q$_{48}$ peptide, and rely on a non-exhaustive docking procedure that delivers a few, but not all possible structures of chaperone/HTT complexes. DNAJB1 might undergo more profound conformational changes upon binding to HTTExon1Q$_{48}$, and the folding and fibrillization mechanisms of HTT

**Fig. 6 | Monomerized DNAJB1 shows increased suppression of HTTExon1Q$_{48}$ aggregation together with Hsc70 and Apg2. a** Experimental validation of DNAJB1$^{wt}$ dimer formation and the monomerized state of DNAJB1$^{\Delta DD}$ that lacks the dimerization domain (DD) by size-exclusion chromatography. The expected molecular weight (MW) of the dimeric DNAJB1$^{wt}$ is 75.96 kDa, and the MW of monomeric DNAJB1$^{\Delta DD}$ is 36.02 kDa. The elution volumes of MW standards Con-albumin and Ovalbumin are indicated and were used for equilibration and to estimate the MW of the purified DNAJB1$^{wt}$ (86.27 kDa) and DNAJB1$^{\Delta DD}$ (32.28 kDa).

**b–d** Analysis of Hsc70, Apg2 and DNAJB1$^{wt}$ or DNAJB1$^{\Delta DD}$ on (**b**) HTTExon1Q$_{48}$ aggregation, (**c**) refolding of heat-denatured luciferase, and (**d**) Aβ$_{1-42}$ fibrilization. The graphs on the left are representative results of three independent experiments. The bar graphs on the right correspond to one-way ANOVA analyses of (**b**) the half-life ($T_{1/2}$) of HTTExon1Q$_{48}$ aggregation, (**c**) the recovered enzymatic activity of refolded luciferase after 2 h, and (**d**) the final fluorescence intensity of ThT, under the tested conditions. Bars represent the mean value, and error bars correspond to the mean SD. ****$P \le 0.0001$; ***$P \le 0.001$; **$P \le 0.01$; ns not significant.

itself should be a subject of further, more advanced computational studies.

In vivo, a number of different chaperones and chaperone complexes within the proteostasis network can attend to HTT to prevent and reverse pathological aggregation. It will be interesting to elucidate how the different chaperones and proteolytic machines are coordinated in a multicellular organism with distinct cell types and with the progression of aging when chaperone capacity is limited due to increased protein damage[32,33]. The different binding sites, but also the interaction with distinct HTT folding states suggest different modes of action exerted by the HTT-binding chaperones. DNAJB6/8 for instance binds to the polyQ stretch and acts in an Hsc70 independent manner[14], whereas we observed no chaperone activity toward HTTExon1Q$_{48}$ for DNAJB1 alone[18]. Consequently, mutation of the HPD of the J-domain of DNAJB1 that is essential for the stimulation of the ATPase activity of Hsc70 (DNAJB1$^{H32Q}$), abrogates the ability of the chaperones to suppress HTTExon1Q$_{48}$ fibrilization (Supplementary Fig. 6). However, DNAJB1 together with Hsc70 and Apg2 constitute a very powerful chaperone complex that can prevent and reverse the aggregation of HTTExon1Q$_{48}$. The identification of the PRD as binding site for DNAJB1 helps in understanding how the same chaperone complex can tackle two very different folding entities of HTT, the soluble monomer as well as the fibrillar state. The prolines have been proposed to form highly flexible structures in HTT fibrils pointing away from the amyloid core[34]. Indeed, we could show that mutating the HBM motif does not only abrogate the ability to prevent HTT aggregation it also strongly reduces the association of DNAJB1 to HTTExon1Q$_{48}$ fibrils. Consequently, the trimeric chaperone complex Hsc70, Apg2, and mutated DNAJB1 (DNAJB1$^{H244A/F}$) is severely impaired in its ability to disaggregate HTTExon1Q$_{48}$ fibrils.

Based on our findings, we propose the following mechanism for the suppression of HTT aggregation and its disaggregation by Hsc70, DNAJB1, and Apg2 (Fig. 10). In Fig. 10a, the model for the suppression of HTT aggregation by Hsc70, DNAJB1, and Apg2 is depicted. Monomeric soluble HTTExon1Q$_{48}$ (HTT) undergoes structural changes, forming early aggregation-competent species (1). DNAJB1 binds to the PRD of HTT (2), using its HBM region, located between the CTDI and CTDII (Fig. 4). H244 is a key residue of the hinge region for this interaction. DNAJB1's CTDII and G/F-rich domain also contribute to HTT binding and suppression of aggregation (Fig. 7b, Supplementary Fig. 5c). Through the established interaction, DNAJB1 targets HTT to Hsc70 (3). As dimer, DNAJB1 is able to bind one HTT protein per protomer and also to recruit two Hsc70 proteins in parallel[35] (Fig. 5c). The binding of Hsc70 to DNAJB1 stimulates the ATPase activity of Hsc70, thus this interaction triggers the Hsc70 chaperone cycle. Once this ternary complex is formed, HTT is transferred from DNAJB1 to Hsc70, following DNAJB1 dissociation (4). When DNAJB1 dissociates, the interaction between Hsc70 and HTT is possibly mediated by its P2 region, which binds a highly positive amino acid patch located in the NBD of Hsc70 (Fig. 1a), and by its N17 domain, which has been shown to interact with the substrate binding domain (SBD) of Hsc70[15]. The dual binding of HTT by Hsc70 could exert a stretching force on HTT facilitated by the putative distal location of HTT N17 and P2 regions (Supplementary Fig. 3b) and the substantial rearrangement of the

NBD and SBD upon ATP hydrolysis and substrate binding[36,37]. At last, Apg2 intervenes to facilitate ADP-ATP exchange in Hsc70, which in turn re-adopts its open conformation and releases HTT (5). Once released, HTT could be rebound by DNAJB1 and the chaperone cycle could be repeated (6). Our observation that ATP is the limiting factor for aggregation suppression supports the assumption of continuous chaperone cycles (step 1–6) in order to suppress HTT aggregation[18]. In addition, other interaction partners, including chaperones such as DNAJB6, NAC, MOAG-4, TriC could also interact with HTT and affect its conformational state[11,12,15,38].

Figure 10b depicts the model for binding and disaggregation of HTT fibrils by Hsc70, DNAJB1 and Apg2. HTT fibrils consist of a core formed by polyQ β-hairpins (Fig. 10b, left panel). The polyQ flanking regions (N17 and PRD) are flexible, dynamic and exposed to the environment[39,40]. The observation that the anti-HTT antibody MW7 can bind its epitope at the intersection of the polyQ core and the PRD in HTT fibrils, further supports the assumption that the P2 region is accessible for binding by DNAJB1[41]. This interaction depends also on the HBM of DNAJB1. DNAJB1 could bind HTT fibrils at multiple positions as previously reported for α-synuclein: (a) at fibril ends[31,42], (b) along the surface of mature fibrils[22,43]", and (c) to short fibrils and oligomers[44] derived from secondary nucleation processes. For HTT, we have previously observed chaperone binding along the fibril and not only at the ends of fibrils formed by HTTExon1Q$_{48}$[18]. Along the fibrils, DNAJB1 interacts with the P2 region of HTT (1) and once bound, recruits Hsc70 to the fibrils, inducing its ATPase activity (2). Upon interaction of DNAJB1 with HTT and Hsc70, DNAJB1 transfers HTT to Hsc70 and returns in its free-state, available for the next cycle of substrate binding (3). As the polyQ domain of HTT adopts a β-hairpin structure in the fibrils, the N17 and PRD domains are in close proximity. Hsc70 could use its dual binding mode (see above) to bind the N17 and the PRD of the same or adjacent HTT proteins. The allosteric changes of Hsc70 NBD and SBD upon ATP hydrolysis could exert forces that destabilize the HTT fibrils. As above mentioned, Apg2 facilitates ADP-ATP exchange in Hsc70, which in turn re-adopts its open conformation and releases HTT (4). Depending on the HTT species initially bound by the chaperones, the process could release fragments, oligomers or monomers of HTT that are then further disaggregated (repeated cycles of steps 1–4) or suppressed in their aggregation propensity (as outlined above).

The identification of a distinct binding site within DNAJB1 for HTT advances our understanding how a generalist such as DNAJB1 can specifically recognize and target a specific substrate to Hsc70. Even though H244 of the HBM is crucial for the interaction of DNAJB1 with soluble HTT as well as HTT amyloid fibrils, there are likely mechanistic differences between binding and targeting of either soluble or aggregated HTT to Hsc70. The contribution of the G/F-rich region as well as the role of the dimerization of DNAJB1 differ between chaperoning of soluble vs aggregated HTT. This raises the question how the binding and chaperone activity of DNAJB1 and in particular as part of a chaperone cycle together with Hsc70 and Apg2 differs along the folding pathway of HTT. The intrinsically disordered nature of the G/F-rich region of DNAJB1 could play a role in the transition from soluble, to oligomeric conformers, to aberrant HTT condensates that may precede the formation of HTT amyloid fibrils[45].

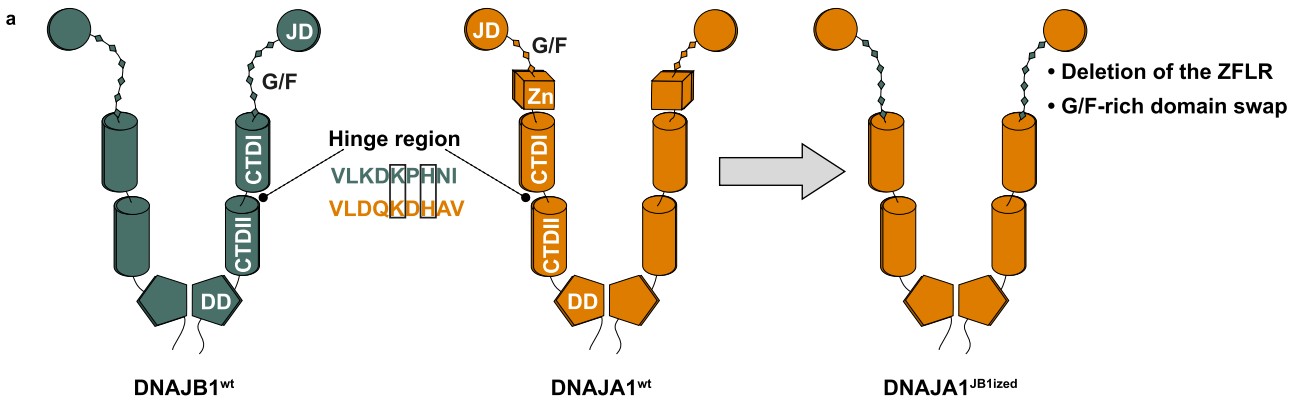

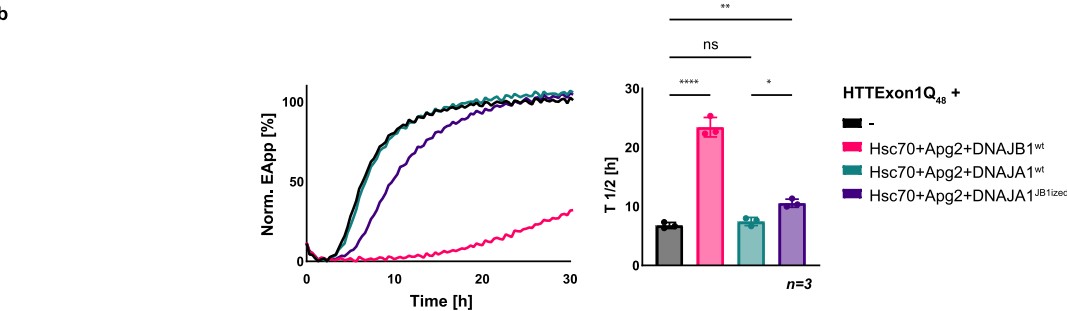

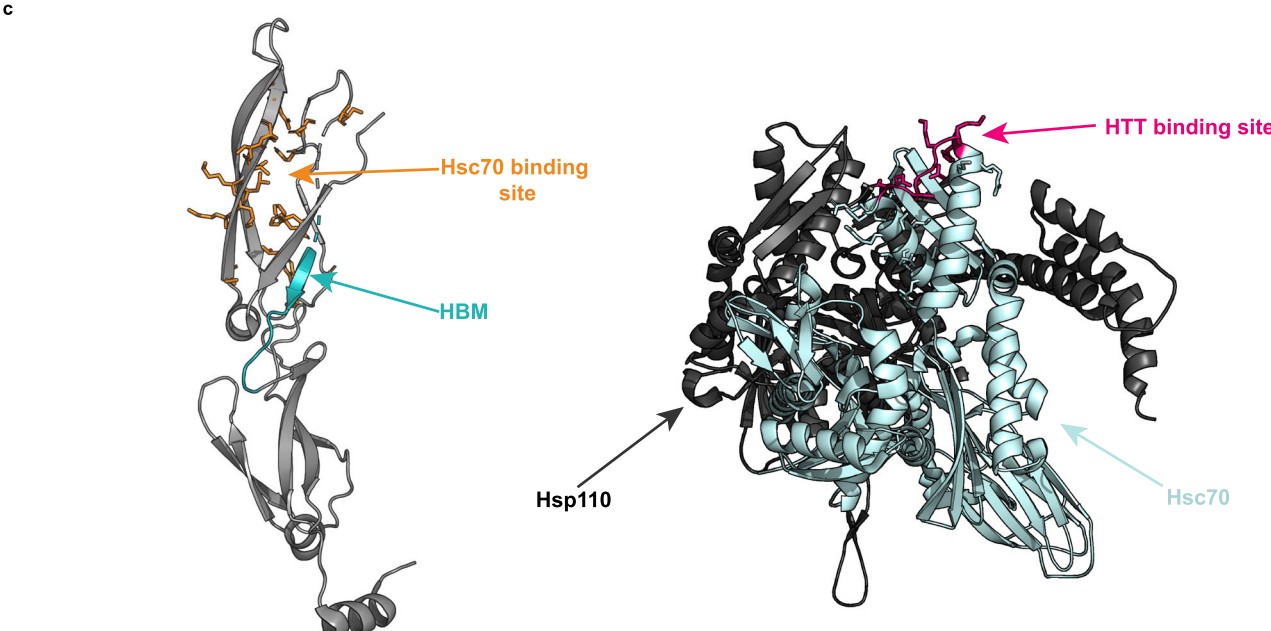

**Fig. 7 | DNAJB1 G/F-rich domain contributes to the suppression of HTTEx-on1Q$_{48}$ aggregation. a** Schematic representation of class B J-domain protein DNAJB1$^{wt}$ (in teal) and class A J-domain protein DNAJA1$^{wt}$ (in orange) and their domains. DNAJA1 possesses a similar HBM in its hinge region as DNAJB1. H and K residues are conserved (black squares). To create DNAJA1$^{JB1ized}$, DNAJA1 ZFLR was deleted, and the short G/F-rich domain of DNAJA1 was substituted by the longer G/F-rich domain of DNAJB1. **b** FRET assay of HTTExon1Q$_{48}$ aggregation and the effect of the addition of Hsc70, Apg2, and DNAJB1$^{wt}$, DNAJA1$^{wt}$ or variant DNAJA1$^{JB1ized}$. The graph is a representative result of three independent experiments, and a one-way

ANOVA analysis of the half-life (T$_{1/2}$) of HTTExon1Q$_{48}$ aggregation under the tested conditions is depicted on the right. Bars represent the mean value and error bars correspond to the mean SD. ****$P \leq 0.0001$; **$P \leq 0.01$; *$P \leq 0.05$; ns not significant. **c** Left, the crystal structure of DNAJB1 protomer (pdb: 3agz) showing in orange the amino acids identified as the binding site for Hsc70[23], and in teal the huntingtin binding motif (HBM) identified in our study. Right, crystal structure of Hsp110-Hsc70 complex (pdb: 3c7n). Hsp110 is displayed in dark gray, and Hsc70 is shown in light blue. The huntington binding site of Hsc70 identified by our cross-linking-MS experiments (Fig. 1a) is shown in magenta.

## Methods

### Cross-linking and mass spectrometry

HTT and chaperone proteins were thawed on ice and transferred to centrifugation tubes (Microfuge Tube Polypropylene, Beckman Coulter). Ultracentrifugation (Optima MAX Ultracentrifuge, Beckman) was carried out at 50000 rpm for 1 h at 4 °C. Protein supernatants (around 95% of the initial volume) were transferred to low-binding tubes, and molar concentration was determined via Bradford assay. Samples containing 5–10 µM total protein and 0.5–1 mM SDA were incubated for 30 min at room temperature. Cross-linking was stopped by adding 1 M Tris-base, samples were then put on ice and photoactivated for 30 min under UV light. SDA cross-linked proteins were loaded on SDS-PAGE and subjected to in-gel digestion. In brief, gel bands were reduced with 5 mM DTT at 56 °C for 30 min and alkylated with 40 mM chloroacetamide at room temperature for 30 min in the dark. Protein digestion was carried out using elastase at an enzyme-to-protein ratio of 1:20 (w/w) at 37 °C overnight.

### LC/MS and data analysis

LC/MS analysis was performed using an UltiMate 3000 RSLC nano LC system coupled online to an Orbitrap Fusion mass spectrometer (ThermoFisher Scientific). Reversed-phase separation was performed using a 50-cm analytical column (in-house packed with Poroshell 120 EC-C18, 2.7 µm, Agilent Technologies) with a 80 min gradient. Cross-link acquisition was performed using a LC-MS2 method with both HCD and ETD fragmentation. The following parameters were applied: MS resolution 120.000; MS2 resolution 60.000; charge state 4–8 enable for MS2. Data analysis was performed using XlinkX[46] with the following parameters: unspecific in silico digestion with minimum peptide length = 6; maximal peptide length = 30 and missed cleavages = 3; fix modification: Cys carbamidomethyl = 57.021 Da; variable modification: Met oxidation = 15.995 Da; SDA cross-linker = 82.04186; precursor mass tolerance = 10 ppm; fragment mass tolerance = 20 ppm. MS2 spectra were searched against a database containing HTTExon1Q$_{48}$ and DNAJB1 proteins.

### Plasmids

pGEX-6P1-HttExon1Q$_{48}$-CyPet, and pGEX-HttExon1Q$_{48}$-YPet were obtained as described previously[18]. pGEX-6P1-HttExon1Q$_{48}$ΔN17-CyPet/YPet were previously generated[8]. pET-6His-Smt3-Apg2, pET-6His-Smt3-Hsc70, and pET-6His-Smt3-DNAJB1 were obtained from the Bukau lab[47]. From Addgene, we obtained: pET-Sac-Abeta(M-42) (from Dominic Walsh, #71875), pcDNA5/FRT/TO V5 DNAJA1 (from Harm Kampinga, #19518), mRFP-FKBP12 (from Tamas Balla, #67514) and pGEX-2T-FRB (from Jie Chen, #26607). pCDNA3-HTTExon1Q$_{97}$-EGFP and pCDNA3-DNAJB1 were obtained as described previously[18].

### Plasmid cloning

For the expression of DNAJA1, the gene was amplified by PCR from pcDNA5/FRT/TO V5 DNAJA1 plasmid using primers (A-B) with BamHI and XhoI restriction sites for further enzymatic digestion and insertion into pET-6His-Smt3 plasmid by ligation.

Deletion variants HTTExon1Q$_{48}$ΔP2-CyPet/YPet (deletion of residues 92–106 with primers C and D), DNAJB1$^{ΔDD}$ (deletion of residues 324–340 with primers E-F), DNAJA1$^{ΔZFLR}$ (deletion of residues 121–205 with primers G-H) and DNAJB1$^{ΔG/F}$ (deletion of residues 89–154 with primers I-J) were obtained by inverse PCR using phosphorylated primers and plasmid recircularization by ligation. HTTExon1Q$_{48}$ΔP2-CyPet/YPet constructs were generated previously[8].

Plasmids for expression of DNAJB1 variants H244A (K-L), H244R (M-N), H244F (U1-V1), H244Q (W1-X1), H32Q (Y1-Z1), H32Q-H244A (Y1-Z1 on DNAJB1$^{H244A}$ as template), K242A (O-P), K242R (Q-R), D234A (S-T), F247A (U-V), R249A (W-X), E173A, and E174A were obtained by QuikChange™ site-directed mutagenesis PCR (Stratagene, La Jolla, CA).

PCR products were cleaned (NucleoSpin Gel and PCR Clean-up, Macherey Nagel) before transformation into *E. coli* (NEB, #C3040H).

For DNAJA1$^{JB1ized}$ and DNAJB1$^{JA1-G/F}$ cloning, DNAJA1$^{ΔZFLR}$ and DNAJB1$^{wt}$ plasmids were used as templates using the ABC cloning protocol described previously[48]. The G/F-rich domains of DNAJB1 (residues 70–164) and DNAJA1 (residues 70–106) were amplified (PCR 1) using primers C1-D1 and G1-H1, respectively. DNAJA1 G/F-rich domain was inserted between DNAJB1 E69 and V164, and DNAJB1 G/F-rich domain was inserted between DNAJA1 E69 and V107. Next, DNAJB1 and DNAJA1$^{ΔZFLR}$ were deleted from their G/F-rich domains by inverse PCR (PCR 2) using primers E1-F1 and I1-J1. The DNA products obtained from PCR 1 and PCR 2 were separated in an agarose gel and cleaned (QIAquick Gel Extraction Kit, Qiagen). Clean fragments were combined in a third PCR reaction where primers C1-F1 were used to assemble DNAJB1 to DNAJA1 G/F-rich domain, and primers G1-J1 to assemble DNAJA1$^{ΔZFLR}$ to DNAJB1 G/F-rich domain. PCR products were cleaned (NucleoSpin Gel and PCR Clean-up, Macherey Nagel) and transformed into *E. coli* (NEB, #C3040H).

For DNAJB1$^{ΔDD}$-FKBP12, DNAJB1$^{ΔDD}$-FRB, DNAJB1$^{ΔDD-H244A}$-FKBP12 and DNAJB1$^{ΔDD-H244A}$-FRB cloning, FKBP12 (K1-L1) and FRB (O1-P1) were amplified from mRFP-FKBP12 and pGEX-2T-FRB plasmids (PCR 1), respectively. Forward primers contained the GGGS linker (DNA sequence GGTGGAGGCGGTTCA) as overhang and reverse primers contained a XhoI restriction site as overhang. DNAJB1$^{ΔDD}$ (M1-N1) and DNAJB1$^{ΔDD-H244A}$ (M1-R1) were amplified using pET-His-Smt3-DNAJB1$^{wt}$ and pET-His-Smt3-DNAJB1$^{H244A}$ as templates (PCR 2), respectively. Forward primers contained a BamHI restriction site as overhangs and reverse primers contained the GGGS linker as overhang. PCR 1 and PCR 2 products were separated in an agarose gel and purified (QIAquick Gel Extraction Kit, Qiagen). A third PCR, using primers M1-L1 for DNAJB1$^{ΔDD}$-FKBP12 and DNAJB1$^{ΔDD-H244A}$-FKBP12, and primers M1-P1 for DNAJB1$^{ΔDD}$-FRB and DNAJB1$^{ΔDD-H244A}$-FRB, was performed to assemble PCR 1 and PCR 2 products together. After agarose gel separation and purification (QIAquick Gel Extraction Kit, Qiagen) PCR 3 products were digested with BamHI and XhoI enzymes and ligated to the pSUMO-His-Smt3 plasmid.

For pCDNA3-DNAJB1$^{H244A}$, DNAJB1$^{H244}$ sequence was amplified by PCR from pSUMO-His-DNAJB1$^{H244A}$ using primers with BamHI and XhoI restriction sites as overhangs (S1-T1). PCR product and an empty pCDNA3 vector were then digested with BamHI and XhoI restriction enzymes and ligated.

### Oligonucleotides used for molecular cloning

| Construct | Ref | Sense | Sequence |
|---|---|---|---|
| DNAJA1 | A | Fw | accggatccaccaatggtgaaagaaacaacttactacg |
|  | B | Rv | tgctcgagcggccgcttaagaggtctgacactgaacacc |
| HTTExon1Q$_{48}$ΔP2-CyPet/YPet | C | Fw | gcggctgaggagccgctgcaccg |
|  | D | Rv | ctgaggcagcagcggctgtgcctg |
| DNAJB1$^{ΔDD}$ | E | Fw | taggctcgagcaccaccaccaccaccactgagatc |
|  | F | Rv | ggggaagatcacttcaaactcaataatgaggtcc |
| DNAJA1$^{ΔZFLR}$ | G | Fw | ggaaagagcatagttcgagagaagaaaatttttag |
|  | H | Rv | gtcggggttaatcacattctttgtcagagccag |
| DNAJB1$^{ΔG/F}$ | I | Fw | gagcccgcccgaaagaagcaagatc |
|  | J | Rv | agaggtaccattggcaccaccgccg |
| DNAJB1$^{H244A}$ | K | Fw | gttttaaaggacaagcccgccaatatctttaagagagatg |
|  | L | Rv | catctctcttaaagatattggcgggcttgtcctttaaaac |
| DNAJB1$^{H244R}$ | M | Fw | gttttaaaggacaagccccgcaatatctttaagagagatg |
|  | N | Rv | catctctcttaaagatattgcggggcttgtcctttaaaac |
| DNAJB1$^{H244F}$ | U1 | Fw | gtctttgtttaaaggacaagcccttcaatatctttaagagagatggctc |
|  | V1 | Rv | gagccatctctcttaaagatattgaagggcttgtcctttaaaacaaagac |
| DNAJB1$^{H244Q}$ | W1 | Fw | gttttaaaggacaagccccagaatatctttaagagagatgg |
|  | X1 | Rv | ccatctctcttaaagatattctggggcttgtcctttaaaac |

| Construct | Primer | Fw/Rv | Sequence |
|---|---|---|---|
| DNAJB1$^{H32Q}$ | Y1 | Fw | caggcgctgcgctaccaaccggacaagaacaag |
| | Z1 | Rv | cttgttcttgtccggttggtagcgcagcgcctg |
| DNAJB1$^{K242A}$ | O | Fw | gtctttgttttaaaggacgcgccccacaatatctttaag |
| | P | Rv | cttaaagatattgtggggcgcgtcctttaaaacaaagac |
| DNAJB1$^{K242R}$ | Q | Fw | cgtctttgtttaaaggaccggccccacaatatctttaagag |
| | R | Rv | ctcttaaagatattgtggggccggtccttaaaacaaagacg |
| DNAJB1$^{D234A}$ | S | Fw | ccaacaacattccagctgctatcgtctttgtttttaaag |
| | T | Rv | ctttaaaacaaagacgatagcagctggaatgttgttgg |
| DNAJB1$^{F247A}$ | U | Fw | caagccccacaatatgctaagagagatggctctg |
| | V | Rv | cagagccatctctcttagcgatattgtggggcttg |
| DNAJB1$^{R249A}$ | W | Fw | cacaatatctttaaggcagatggctctgatgtc |
| | X | Rv | gacatcagagccatctgccttaaagatattgtg |
| DNAJB1$^{E173A}$ | Y | Fw | cttcgagtctcccttgcagagatctacagcggc |
| | Z | Rv | gccgctgtagatctctgcaagggagactcgaag |
| DNAJB1$^{E174A}$ | A1 | Fw | gagtctcccttgaagcgatctacagcggctg |
| | B1 | Rv | cagccgctgtagatcgcttcaagggagactc |
| DNAJB1$^{JA1-G/F}$ | C1 | Fw1 | gaccgctacggggagcaggcaattaaagagggtgg |
| | D1 | Rv1 | aaggtcgtgggtgacatttttacctctcctttctc |
| | E1 | Fw2 | aggagaggtaaaaatgtcacccacgaccttcgagtc |
| | F1 | Rv2 | ctctttaattgcctgctccccgtagcggtcgaagatc |
| DNAJA1$^{JB1ized}$ | G1 | Fw1 | gacaaaggaggagaagaaggcctaaaggggagtgg |
| | H1 | Rv1 | gagctgatgtacaactggggggatcttgcttctttc |
| | I1 | Fw2 | gaaagaagcaagatccccagttgtacatcagctctcagtaaccc |
| | J1 | Rv2 | ccactcccctttaggccttcttctcctcctttgtcatataattcc |
| DNAJB1$^{ΔDD}$-FKBP12/ DNAJB1$^{ΔDD-H244A}$-FKBP12 | K1 | Fw1 | cccggtggaggcggttcaggagtgcaggtggaaccatc |
| | L1 | Rv1 | ccgctcgagtcattccagttttagaagctc |
| | M1 | Fw2 | taagcaggatccaatgggtaaagactactac |
| | N1 | Rv2 | tcctgaaccgcctccaccgggggaagatcacttcaaactc |
| DNAJB1$^{ΔDD}$-FRB/ DNAJB1$^{ΔDD-H244A}$-FRB | O1 | Fw1 | cccggtggaggcggttcaatcctctggcatgagatgtgg |
| | P1 | Rv1 | ccgctcgagtcacttttgagattcgtcggaacac |
| | Q1 | Fw2 | taagcaggatccaatgggtaaagactactac |
| | R1 | Rv2 | gattgaaccgcctccaccgggggaagatcacttcaaactc |
| pCDNA3-DNAJB1$^{H244A}$ | S1 | Fw | taagcaggatccaatgggtaaagactactac |
| | T1 | Rv | atagtcctcgagctatattggaagaacctgc |

## Protein expression and purification

GST-HTTExon1Q$_{48}$-CyPet/YPet

The amino acid sequence of HTTExon1Q$_{48}$ is:

MATLEKLMKAFESLKSFQQQQQQQQQQQQQQQQQQQQQQQQQ
QQQQQQQQQQQQQQQQQQQQQQQQPPPPPPPPPPPPQLPQPPPQAQP
LLPQLQPPPPPPPPPPGPAAAEEPLHRP

In total, 50 ng of pGEX-HTTExon1Q$_{48}$-CyPet/YPet plasmids were transformed into *E. coli* BL21-CodonPlus (DE3)-RIPL cells (Agilent, #230280). A starter culture was prepared by inoculating 30 ml of LB supplemented with Ampicillin (100 µg/ml) media with 5 transformants followed by incubation at 37 °C and shaking at 180 rpm overnight. The next morning, 1 L of LB-Ampicillin media were inoculated with 10 ml of the starter culture and incubated at 37 °C and shaking at 180 rpm. When OD$_{600}$ reached 0.6, protein expression was induced by adding 1 mM IPTG and continuing incubation at 20 °C with shaking at 180 rpm for 4 h. Bacteria were pelleted by centrifugation at 8000 rpm for 20 min at 4 °C. Pellet was resuspended in 100 ml of lysis buffer pH 8.0 (50 mM NaH$_2$PO$_4$, 5 mM Tris, 150 mM NaCl, 1 mM EDTA, 1 mM PMSF, 1 tab/50 ml of cOmplete EDTA-free protease inhibitor cocktail (Roche)) and subjected to four cycles at 16.000 psi of cell disruption (LM10 Microfluidizer, Microfluidics). The lysate was supplemented with 1% Triton X-100 and incubated with stirring for 10 min at room temperature. Next, the soluble fraction was recovered by centrifugation at 16.000 rpm (Sorvall Evolution RC, Thermo Scientific) for 40 min at 4 °C. In total, 2 ml of glutathione sepharose (GE Healthcare #17-0756-05) were added to the cleared lysate and GST-tagged proteins were allowed to bind under gentle rotation for 1 h at 4 °C. Bound slurry was filtered through a gravity flow column and washed with 25 ml of wash buffer pH 8.0 (50 mM NaH$_2$PO$_4$, 5 mM Tris, 150 mM NaCl, 1 mM EDTA, 1 tab/50 ml of cOmplete EDTA-free protease inhibitor cocktail (Roche), 0.1% Triton X-100). GST-tagged protein was recovered after incubation with 4 ml of elution buffer pH 8.6 (50 mM NaH$_2$PO$_4$, 5 mM Tris, 150 mM NaCl, 1 mM EDTA, 20 mM reduced glutathione) for 1 h at 4 °C. Eluted protein was transferred to a 6–8 kDa MWCO dialysis membrane (Spectra/Por 1, Spectrum laboratories) and buffer was exchanged overnight at 4 °C against 2 L of dialysis buffer (50 mM Tris pH 7.4, 150 mM NaCl, 1 mM EDTA, 5% glycerol). Protein was aliquoted and flash-frozen in liquid nitrogen for storage at −80 °C.

**Chaperones.** In all, 50 ng of pET-6His-Smt3 plasmids containing the chaperone sequences were transformed into *E. coli* BL21-CodonPlus (DE3)-RIPL cells (Agilent, #230280). A starter culture was prepared by inoculating 30 ml of LB supplemented with Kanamycin (50 µg/ml) media with 5 transformants followed by incubation at 37 °C and shaking at 180 rpm overnight. The next morning, 2 L of LB-Kanamycin media were inoculated with 20 ml of the starter culture and incubated at 37 °C and shaking at 180 rpm. When OD$_{600}$ reached 0.6, protein expression was induced by adding 1 mM IPTG and continuing incubation at 20 °C with shaking at 180 rpm overnight. The next morning, bacteria were pelleted by centrifugation at 8000 rpm during 20 min at 4 °C. Pellet was thawed in ice and resuspended in 100 ml of lysis buffer (30 mM HEPES pH 7.4, 500 mM KCl, 5 mM MgCl$_2$, 30 mM imidazole, 10% glycerol, 1 mM PMSF, 1 mM β-mercaptoethanol, 10 µg/ml DNase I, 1 tab/50 ml of cOmplete EDTA-free protease inhibitor cocktail (Roche)). The cell suspension was subjected to four cycles of cell disruption at 16.000 psi (LM10 Microfluidizer, Microfluidics) and soluble fraction was recovered after centrifugation at 16.000 rpm (Sorvall Evolution RC, Thermo Scientific) for 40 min at 4 °C. In total, 3 ml of Ni-NTA slurry (High-Density Nickel 6BL-QNi-100, Agarose Bead Technologies) were added to the soluble fraction, and His-tagged proteins binding was allowed to occur at 4 °C with gentle rotation over a period of 1 h. Slurry was filtered through a gravity flow column and washed with 25 ml of high-salt (30 mM HEPES pH 7.4, 1 M KAc, 5 mM MgCl$_2$, 25 mM imidazole, 10% glycerol, 1 mM β-mercaptoethanol) and low-salt (30 mM HEPES pH 7.4, 50 mM KAc, 5 mM MgCl$_2$, 25 mM imidazole, 10% glycerol, 1 mM β-mercaptoethanol) buffers. His-tagged proteins were recovered after addition of 8 ml of elution buffer (30 mM HEPES pH 7.4, 100 mM KAc, 5 mM MgCl$_2$, 300 mM imidazole, 10% glycerol, 1 mM β-mercaptoethanol) and incubation for 30 min at 4 °C with gentle rotation. Elution fraction was transferred to a 6–8 kDa MWCO dialysis membrane (Spectra/Por 1, Spectrum laboratories) and buffer was exchanged overnight at 4 °C against 2 L of dialysis buffer (30 mM HEPES pH 7.4, 100 mM KAc, 5 mM MgCl$_2$, 10% glycerol, 1 mM β-mercaptoethanol) supplemented with 40 µl of 4 mg/ml His-Ulp1 protease to cleave the His-Smt3 tag. To remove the cleaved tag and the protease, protein solution was incubated with 1.5 ml of Ni-NTA slurry for 30 min at 4 °C and filtered through a gravity flow column. The recovered protein was aliquoted and flash-frozen in liquid nitrogen for storage at −80 °C.

**Aβ42.** Aβ42 peptide was overexpressed and purified as described previously[49]. In all, 50 ng of pET-M-Aβ42 plasmid were transformed into *E. coli* BL21-CodonPlus (DE3)-RIPL cells (Agilent, #230280). A starter culture was prepared by inoculating 30 ml of LB supplemented with Ampicillin (100 µg/ml) media with 5 transformants followed by incubation at 37 °C and shaking at 180 rpm overnight. The next morning, 1 L of LB-Ampicillin media was inoculated with 10 ml of the starter culture and incubated at 37 °C and shaking at 180 rpm. When OD$_{600}$ reached 0.6, protein expression was induced by adding 0.5 mM IPTG, and continuing incubation for 4 h followed by centrifugation at 8.000 rpm (Sorvall Evolution RC, Thermo Scientific) for 20 min at 4 °C. Pellet was resuspended in 100 ml of lysis buffer (50 mM Tris-HCl pH 8.0, 100 mM NaCl, 1% NP40, 2 mM PMSF and 1 tab/50 ml of cOmplete

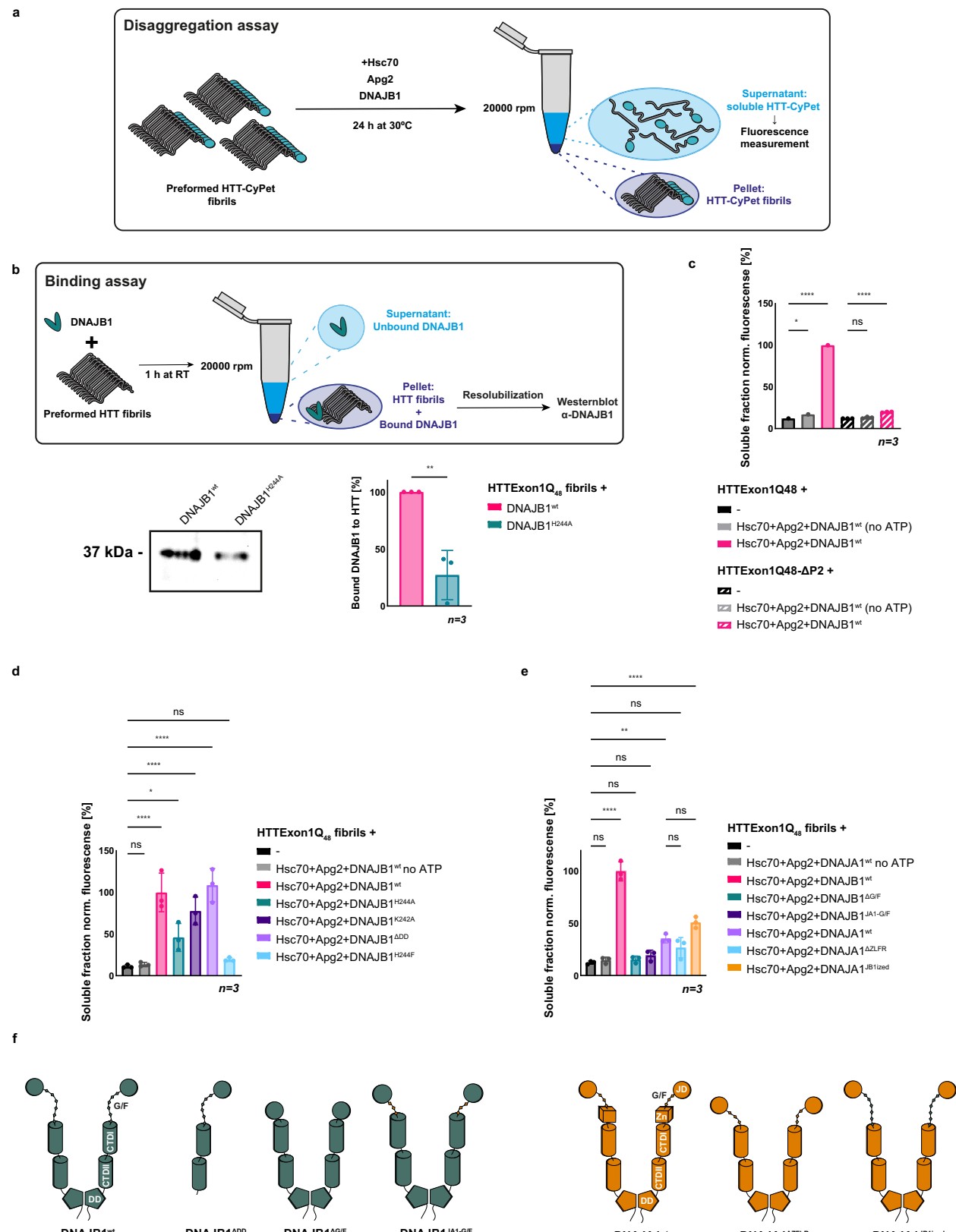

EDTA-free protease inhibitor cocktail (Roche)), and subjected to 4 cycles of cell disruption at 16000 psi (LM10 Microfluidizer, Microfluidics). The insoluble fraction containing the Aβ42 peptide was recovered by centrifugation at 12.000×*g* (Sorvall Evolution RC, Thermo Scientific) for 30 min at 4 °C. To clean the peptide from cell debris, the insoluble fraction was resuspended in cleaning buffer (lysis buffer supplemented with 300 μg/ml of lysozyme and 2 mM EDTA) and incubated for 1 h at room temperature with gentle rotation. The suspension was next centrifuged at 12.000×*g* for 10 min at 4 °C, and the pellet was washed twice by resuspension in 35 ml of wash buffer 1 (PBS pH 8.0, 0,5% Triton X-100), incubation for 10 min at room temperature with gentle rotation and centrifugation at 12.000×*g* for 10 min. Next,

**Fig. 8 | DNAJB1^H244 is key for binding and disaggregation of HTTExon1Q48 fibrils.**
**a** Schematic representation of the disaggregation assay. HTTExon1Q48-CyPet fibrils were incubated with the indicated chaperones and ATP for 24 h. The resolubilized moiety was separated from fibrils by centrifugation, and CyPet fluorescence of the supernatant was measured as a readout of the resolubilized HTTExon1Q48. **b** Top, schematic representation of the HTT fibril-binding assay. Preformed HTTExon1Q48 fibrils were incubated with DNAJB1^wt or DNAJB1^H244A and their association with the fibrils assessed by a sedimentation analysis and subsequent Western blot. Bottom, Western blot of the intensities of DNAJB1 and DNAJB1^H244A bound to HTTExon1Q48 fibrils. Quantification of three independent experiments is depicted on the right.

Bars represent the mean value, and error bars correspond to the mean SD. **P ≤ 0.01. **c–e** Fluorescence level measurements of three independent experiments of disaggregated HTTExon1Q48 or HTTExon1Q48ΔP2 (only **c**) by Hsc70, Apg2 and DNAJB1^wt (**c**) or variants DNAJB1^H244A, DNAJB1^K242A, DNAJB1^H244F, DNAJB1^ΔDD (**d**) or DNAJB1^ΔG/F, DNAJB1^JAI-G/F, DNAJA1^wt, DNAJA1^ΔZFLR, and DNAJA1^JBlized (**e**). The significance was determined by a one-way ANOVA analysis. Bars represent the mean value and error bars correspond to the mean SD. ****P ≤ 0.0001; ***P ≤ 0.001; **P ≤ 0.01; *P ≤ 0.05; ns not significant. **f** Schematic representation of the DNAJB1 and DNAJA1 variants analyzed in (**e**).

two wash steps were performed in 35 ml of wash buffer 2 (PBS pH 8.0), followed by ten cycles of 10 s sonication at 50% amplification (HD 2070, Bandelin Sonoplus). The suspension was divided into four equal volumes and centrifuged at 12.000×g for 10 min at 4 °C. In order to solubilize the peptide, each pellet was resuspended in 35 ml of 50 mM NaOH and incubated for 2 h at room temperature with gentle rotation, followed by a step of centrifugation at 12.000×g for 10 min at 4 °C. The supernatant was recovered, and its pH was carefully adjusted to 7.0 by slow addition of 0.5 M HCl. The suspension was centrifuged at 12.000×g for 10 min at 4 °C, and the supernatant containing the soluble Aβ42 was recovered. The remaining pellet was subjected to another round of NaOH solubilization and pH neutralization. The recovered supernatants were pooled and filtered through a 30 kDa MWCO filter (Amicon Ultra-15 Centrifugal filters, Millipore). The flow-through was next concentrated to around 4–5 ml in a 3 kDa MWCO filter (Amicon Ultra-15 Centrifugal filters, Millipore), and 1 ml aliquots were prepared and flash-frozen in liquid nitrogen for storage at −80 °C.

### Biochemical assays
Prior to all biochemical assays, proteins were thawed on ice and transferred to centrifugation tubes (Microfuge Tube Polypropylene, Beckman Coulter). Ultracentrifugation (Optima MAX Ultracentrifuge, Beckman) was carried out at 40.000 rpm for 40 min at 4 °C. Protein supernatants (around 95% of the initial volume) were transferred to low-binding tubes (Sorenson) and molar concentration was determined via Bradford assay.

### FRET-based assay for HTTExon1Q48 fibrilization analysis
This assay was performed according to previously published protocols[18]. Briefly, 100 µl samples were prepared in low-binding tubes containing 30 mM HEPES pH 7.4, 150 mM NaCl, 5 mM MgCl2, 1 mM DTT, 1 µl/50 µl of Pyruvate Kinase (Sigma-Aldrich, #P7768), 3 mM Phosphoenol Pyruvate, 5 mM ATP, and equimolar concentrations of GST-HTTExon1Q48-CyPet and GST-HTTExon1Q48-YPet to make a final concentration of 0.75 µM. If stated, chaperones were added at concentrations of Hsc70 5 µM, Apg2 0.25 µM, and J-domain protein 5 µM or 10 µM for 2× conditions. For the analysis of DNAJB1 chemically induced dimers (DNAJB1-CIDs), FKBP12 and FRB-fused pairs were added at 2.5 µM to make a total of 5 µM of CID and Rapamycin (Cayman Chemical, #Cay13346) concentration was 100 µM. In addition, donor and acceptor samples were prepared containing only 0.75 µM GST-HTTExon1Q48-CyPet or GST-HTTExon1Q48-YPet, respectively. Fluorescence measurements were carried out with the Tecan i-Control TM 1.10.1.0 software of the Infinite F200 PRO Tecan plate reader, and data analyses were carried out as described previously[18]. The half-life time (T_{1/2}) of HTT fibrilization in the absence and presence of all tested condition in this study is depicted in Supplementary Fig. 7.

### Luciferase refolding assay
Luciferase assay was performed according to previously published protocols[47,50]. Briefly, a 15 nM luciferase solution in 1× dilution buffer (50 mM HEPES pH 7.4, 100 mM KCl, 5 mM MgCl2, 1 mM DTT, 10 µM BSA, 3.5 µM Pyruvate Kinase (Sigma-Aldrich, #P7768), 3 mM Phosphoenol Pyruvate) was denatured at 45 °C for 15 min. Next, luciferase

was diluted to a final concentration of 5 nM in 1× dilution buffer containing 2.5 µM Hsc70, 0.125 µM Apg2, and 2.5 µM J-domain protein. At time 0, 7, 15, 30, 60, 90, and 120 min, 5 µl aliquots from each refolding sample were dispensed to three different wells of a 96-wells half-area white microplate (OptiPlate, Perkin Elmer) containing 50 µl of assay buffer (25 mM glycylglycine, 100 mM KCl, 15 mM MgCl2, 5 mM ATP). In all, 50 µl of 0.25 mM luciferin solution were added to each well and luminescence was measured in a plate reader (Infinite F200 PRO, Tecan). Attenuation was not used, integration time was 1000 ms, and settle time was 0 ms. Values were normalized to the values measured for the native luciferase sample at each time point and data were presented as the percentage of luciferase activity recovered over time.

### ATPase assay
ATPase assay was performed according to previous protocols[18,51]. Briefly, 160 µl samples were prepared containing 1× reaction buffer (50 mM HEPES pH 7.4, 100 mM KCl, 5 mM MgCl2, 0,02% Triton X-100), 1 µM HSC70 and 0.5 µM of the J-domain protein to test. To set the beginning of the reaction ATP was added to each sample so that the final concentration was 1 mM. Samples were then incubated at 20 °C during 2.5 h. 50 µl aliquots of phosphate standards and of each sample were transferred in triplicates to a 96-wells transparent microplate (Sarstedt), followed by 160 µl of green malachite reaction solution (2:1:3 dilution of 0.082% green malachite, 5.7% ammonium molybdate (in 6 M HCl) and water) and 20 µl of 34% sodium citrate. Absorbance at 650 nm was then measured in a plate reader (Infinite F200 PRO, Tecan) as a readout of the free phosphate generated by the ATPase activity of Hsc70. The exact concentration of free phosphate present in each well was calculated using the equation obtained from the phosphate equilibration curve, and normalized to the intrinsic ATPase activity of Hsc70.

### Thioflavin T assay to monitor Aβ1–42 aggregation
After ultracentrifugation, Aβ1–42 was loaded onto a size-exclusion column (Superdex 75 10/300) previously equilibrated with 50 mM NaOH. The peptide, which elutes at around 13 ml, was recovered, concentrated and its molar concentration was determined via Bradford assay. In all, 160 µl samples were prepared containing 30 mM HEPES pH 7.4, 150 mM NaCl, 5 mM MgCl2, 10 µM ThT, 5 µM Aβ1–42, and chaperones at 5 µM Hsc70, 0.25 µM Apg2 and 5 µM J-domain protein. In total, 50 µl aliquots of each sample were transferred into three different wells of a 384-well non-binding black microplate (Corning, #3575). The microplate was sealed with a transparent coverslip and ThT fluorescence was measured every 10 min over a period of 30 h using a plate reader (Infinite F200 PRO, Tecan) set at 37 °C. Excitation and emission wavelengths were 440 nm and 490 nm, respectively, and a 10 s and 2 mm shaking cycle was done before each measurement. ThT fluorescence was plotted as a function of time, and all values were normalized to the sample of Aβ1–42 in the absence of chaperones.

### Differential scanning fluorescence analysis
Overall, 160 µl samples were prepared containing 30 mM HEPES pH 7.4, 150 mM NaCl, 5 mM MgCl2, 2× SYPRO Orange protein gel stain

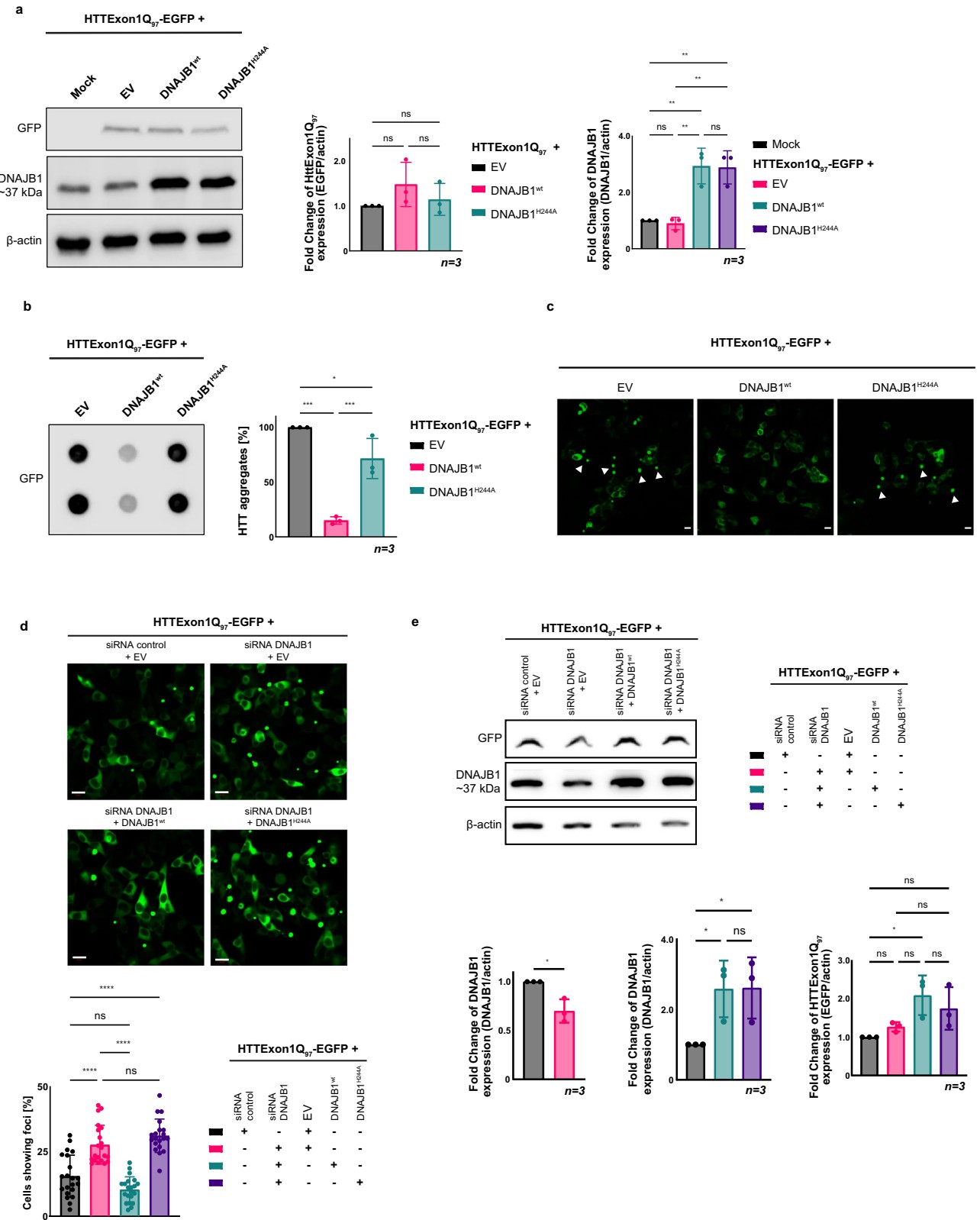

(Sigma-Aldrich, #S5692), and 5 µM of the chaperones. In all, 50 µl aliquots of each sample were transferred into three different wells of a MicroAmp Optical 96-well reaction plate (Applied Biosystems), which was then sealed with a transparent coverslip. A step and hold temperature ramp was performed in a StepOnePlus Real-Time PCR System (Applied Biosystems). First, samples were equilibrated for 2 min at 25 °C. Then, 1 °C increment of temperature jumps were performed

every 2 min from 25–95 °C. The inflexion point of melting curves was considered as the Tm of proteins.

### Size-exclusion chromatography

Superdex 75 10/300 GL and Superose 6 columns connected to an ÄKTA Pure (GE Healthcare) system were used. Columns were equilibrated with Running Buffer (30 mM HEPES-KOH pH 7.4, 50 mM KCl,

**Fig. 9 | Overexpression of DNAJB1$^{wt}$, but not of DNAJB1$^{H244A}$ rescues HTTExon1Q$_{97}$-EGFP aggregation in HEK293 cells. a** Left, western blot analysis of the expression levels of HTTExon1Q$_{97}$-EGFP, DNAJB1, and β-actin after co-transfection of HEK293 cells with HTTExon1Q$_{97}$-EGFP and an empty vector (EV), DNAJB1$^{wt}$ or DNAJB1$^{H244A}$. The depicted blot is representative of three independent experiments. Middle and right, one-way ANOVA analysis of quantification of the overexpression levels of HTTExon1Q$_{97}$-EGFP and DNAJB1$^{wt}$/DNAJB1$^{H244A}$. Bars represent the mean value, and error bars correspond to the mean SD. **$P \le 0.01$; ns not significant. **b** Left, filter-retardation analysis of the aggregation of HTTExon1Q$_{97}$-EGFP in HEK293 cells upon overexpression of an EV, DNAJB1$^{wt}$ or DNAJB1$^{H244A}$. The depicted blot is representative of three independent experiments. Right, one-way ANOVA analysis of the quantification of aggregated HTTExon1Q$_{97}$-EGFP. Bars represent the mean value, and error bars correspond to the mean SD. ****$P \le 0.0001$; ***$P \le 0.001$; *$P \le 0.05$; ns not significant. **c** Confocal images of HEK293 at 48 h after transfection,

expressing HTTExon1Q$_{97}$-EGFP in the presence of an EV, DNAJB1$^{wt}$ or DNAJB1$^{H244A}$. Arrowheads show aggregates. Scale bars are 10 μm. **d** Confocal images of HEK293 cells expressing HTTExon1Q$_{97}$-EGFP and the indicated constructs. A one-way ANOVA analysis of the quantification of the percentage of cells exhibiting HTTExon1Q$_{97}$-EGFP foci is depicted below. Each experiment is represented by a symbol in the respective column and represents a technical replicate that is composed of three biological replicates. For each condition, a total of 703–868 cells were analyzed. Bars represent the mean value, and error bars correspond to the mean SD. ****$P \le 0.0001$; ns not significant. **e** Representative western blot of three independent experiments of the protein levels of HTTExon1Q$_{97}$-EGFP and DNAJB1. A one-way ANOVA analysis of the western blot quantification based on the normalization to β-actin is depicted below. Bars represent the mean value, and error bars correspond to the mean SD. **$P < 0.5$; ns not significant.

1 mM β-mercaptoethanol). Proteins were injected and eluted at a flow rate of 0.5 ml/min in Running Buffer. For the analysis of DNAJB1-CIDSs, equimolar concentrations of the FKBP12 and FRB-fused proteins were added a two times molar excess of Rapamycin and incubated at room temperature for 10 min before loading onto the column.

### HTT fibril-binding assay
In all, 2 μM HTTExon1Q$_{48}$ were aggregated for 4.5 h at 20 °C as described above. DNAJB1$^{wt}$ and DNAJB1$^{H244A}$ were added to HTT fibrils with a final concentration of 1 μM. The samples were incubated 1 h at RT and sedimented at 20.000 rpm for 30 min at 4 °C. The supernatant was carefully removed, and the remaining pellets were resolubilized by incubating with 100% formic acid for 1 h at RT. The formic acid was evaporated by vacuum centrifugation overnight, and the dry pellets were solubilized with 30 mM HEPES, 150 mM KCl buffer and analyzed by Western Blotting.

### Disaggregation assay
HTTExon1Q$_{48}$ fibrils were obtained by preparing samples containing 2 μM GST-HTTExon1Q$_{48}$-CyPet in Aggregation Buffer (30 mM HEPES-KOH pH 7.4, 150 mM KCl, 5 mM MgCl$_2$, 1 mM DTT). To initiate aggregation, PreScission protease was added and incubated for 4.5 h at 20 °C. Fibrillar HTTExon1Q$_{48}$ was next diluted to a final concentration of 0.75 μM in Aggregation Buffer supplemented with 3 mM PEP and 1 μl/50 μl of Pyruvate Kinase (Sigma-Aldrich, #P7768). Chaperones were added at 10 μM Hsc70, 0.5 μM Apg2 and 10 μM J-domain protein. To initiate the disaggregation reaction, 5 mM ATP were added to each sample and allowed to react for 24 h at 30 °C. Then, samples were centrifuged at 20.000×*g* for 30 min at 4 °C and the supernatant was recovered. Fluorescence of the soluble fraction was measured in the Tecan F200 plate reader (Exλ: 430 nm; Emλ: 485 nm) in triplicates and data was normalized to the sample of HTTExon1Q$_{48}$ fibrils with Hsc70, Apg2 and DNAJB1$^{wt}$.

### CD analysis
Proteins were dialyzed overnight in 50 mM phosphate buffer pH 7.4 at 4 °C. For the CD measurements, proteins were diluted in 50 mM phosphate buffer to a final protein concentration of 0.5 mg/ml. CD spectra were recorded with the Applied Photophysics Chirascan spectrometer and evaluation was performed with the Pro-Data Chirascan software (v.4.2.22). Suprasil quartz cells (Hellma UK Ltd.) with a pathlength of 0.1 mm were used. Measurements were performed over a temperature range of 20–90 °C in 5 °C steps, with 5 min equilibration time between each measurement. At least three scans have been performed for each sample over a standard wavelength range of 190–250 nm with intervals of 1 nm. Each protein sample was measured in technical triplicates. Baseline scans have been performed with 50 mM phosphate buffer. The baseline was subtracted from recorded spectra and repetitive scans were averaged before a Savitsky–Golay smoothing filter with smoothing windows of two data points was

applied. Spectra were recorded in raw CD ellipticity ($\Theta$) and were further converted to mean residue ellipticity ($\Theta_{MRE}$):

$$\Theta_{MRE} = \Theta / cln,$$

where c is the protein concentration, l the quartz cuvette pathlength and n the number of amino acids in the protein sequence. For each sample, triplicates were averaged and the standard deviation was determined.

The midpoint of the unfolding transition (Tm) was calculated from the fraction of protein folded at any temperature ($\alpha$) defined as:

$$\alpha = (\Theta T - \Theta U)(\Theta F - \Theta U),$$

where $\Theta T$ is the ellipticity at any temperature, $\Theta U$ is the ellipticity at the unfolded state, and $\Theta F$ at the folded state. $\Theta U$ was defined as the average ellipticity of the two highest temperatures, whereas $\Theta F$ was set as the average ellipticity that was recorded for the two lowest temperatures. Tm is defined as the temperature at which $\alpha = 0.5$[52] and also referred to as the melting temperature. $\Theta T$ values from the different temperature steps at wavelength 194 nm were used in order to calculate $\alpha$. Plotting of temperature against $\alpha$ allows for a sigmoid curve for fitting data to obtain a precise Tm value.

### HEK293 culture, transfection, harvesting, and imaging
HEK293 were obtained from DMSZ (German Collection of Microorganisms and Cell Cultures GmbH #ACC305) and maintained according to standard procedures at 37 °C and 5% CO$_2$, in DMEM + 2 mM glutamate, with 1 g/l glucose (Gibco), supplemented with 10% fetal calf serum (Biochrom), and 100U/ml of both penicillin and streptomycin (Gibco). For overexpression of the HTTExon1Q$_{97}$-EGFP and DNAJB1 constructs, 2 μg of each plasmid were co-transfected using polyethylenimine (Polysciences). 48 h after transfection, cells were either harvested in ice cold PBS and washed twice, or fixed in 4% paraformaldehyde (Sigma) for 10 min at room temperature. Fixed samples were subsequently imaged on a laser scanning confocal microscope Zeiss LSM780 (Carl Zeiss AG), using a Zeiss Plan-Apochromat ×40/1.3 oil DIC M27 objective. For the rescue experiments depicted in Fig. 9x, 250.000 cells were seeded per plate (μ-Dish 35 mm, high from ibidi) 24 h prior to transfection. Cells were transfected with jetPRIME according to the manufacturer's instructions. For expression of HTTExon1Q$_{97}$-EGFP and DNAJB1 constructs or EV (empty vector) controls, 1 μg of each plasmid was used for transfection together with the respective siRNA. The siRNA was purchased from Darmacon (siGENOME SMARTpool Human DNAJB1 (M-012735-02-0005) and siGENOME control pool non-targeting #1 (D-001206-13-05)) and used for transfection at a final concentration of 50 nM. Cells were imaged on a laser scanning confocal microscope (LSM 880, Zeiss) using a Zeiss Plan-Apochromat ×20/0.8 objective 32 h after the transfections.

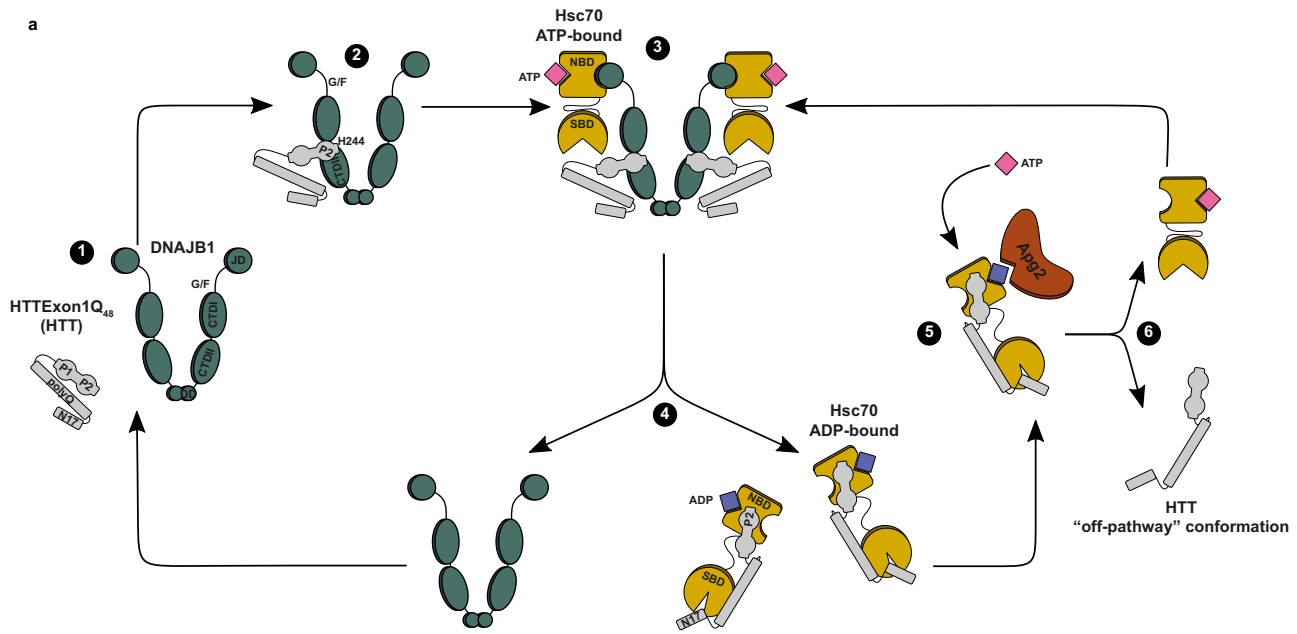

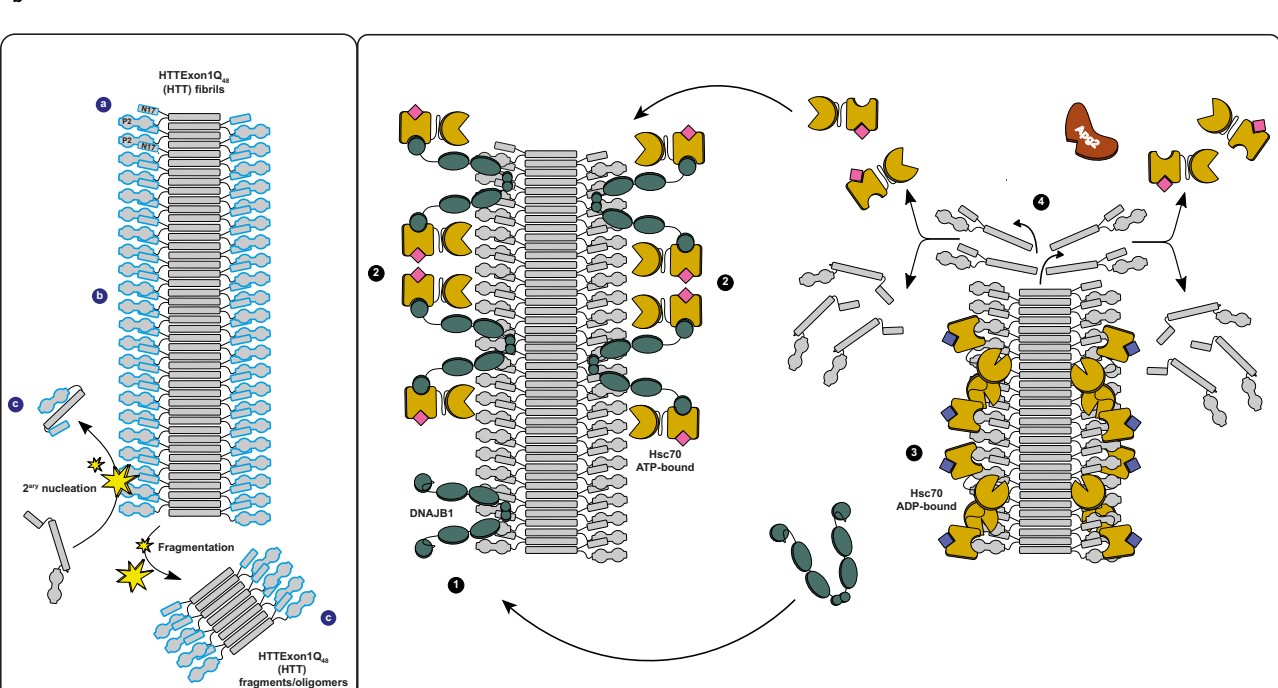

**Fig. 10 | Model of suppression and disaggregation of HTTExon1Q$_{48}$ by the trimeric chaperone complex, Hsc70, DNAJB1, and Apg2. a** Proposed model of the suppression of HTTExon1Q$_{48}$ aggregation by Hsc70, DNAJB1, and Apg2 +ATP. **b** Proposed model of the disaggregation of HTTExon1Q$_{48}$ fibrils by Hsc70, DNAJB1, and Apg2 + ATP. DNAJB1 is depicted in teal, Hsc70 in sand, Apg2 in brown and HTTExon1Q$_{48}$ in gray. The N17, polyQ, PRD with P1 and P2 domains of HTTExon1Q$_{48}$ are indicated on the left. The models are described in "Discussion".

### Western blotting analysis and filter-retardation assay

Western blotting and filter-retardation analysis were performed similarly as previously described[18]. Cell pellets were incubated for 30 min on ice in lysis buffer (50 mM Tris-HCl pH 7.4, 150 mM NaCl, 1 mM EDTA, 1% Triton X-100, and 1 tab/50 ml of complete protease cocktail inhibitor (Roche)). Lysates were centrifuged at 4 °C for 20 min at 20.000×$g$ and the soluble and insoluble fractions were separated and used, respectively, for Western blotting and filter-retardation assay. For

Western blot analysis, 50 μg of total cell lysate per sample were boiled for 10 min at 100 °C in 1× Laemmli buffer (125 mM Tris-HCl pH 6.8, 20% glycerol, 4% SDS, 10% β-mercaptoethanol, 0.004% bromophenol blue), fractionated by SDS-PAGE analysis on a 10% SDS gel in Tris-glycine buffer and transferred onto a nitrocellulose membrane via semi-dry blotting (Trans-Blot Turbo System, BioRad). Western blot membranes were blocked with 5% milk in TBST for 1 h at room temperature and then incubated with the primary antibodies anti-actin (1:4000, Sigma),

anti-DNAJB1 (1:2000, Proteintech), and anti-EGFP (1:1000, Enzo) for detection of HTTExon1Q$_{97}$-EGFP. Membranes were subsequently probed with secondary antibodies against mouse-HRP (1:10.000, ThermoFisher) and rabbit IRDye 680 (1:10.000, LI-COR Biosciences). For filter-retardation analysis, we followed a previous protocol[18]. Blotted membranes were incubated with primary antibody anti-EGFP (1:1000, Enzo) and probed with the secondary anti-mouse-HRP (1:10.000, ThermoFischer). Western blot and filter-retardation assay signals were developed using Pierce ELC (ThermoFisher) and detected on an OdysseyFC imaging system (Image Studio TM software 4.0; LI-COR Biosciences). Image analysis was performed using the Fiji/ImageJ[53] and statistical analysis was conducted with Prism 8 software (GraphPad).

## Protein alignments and conservation

Amino acid sequences were aligned using Clustal Omega, and the resulting file was formatted in the BoxShade server (https://embnet. vital-it.ch/software/BOX_form.html). The graphic representations of amino acids conservation were created using the WebLogo application (https://weblogo.berkeley.edu/). Maestro Software Suite Version 12.7 was used to build protein homology models.

## Classical molecular dynamics (MD) simulation

The crystal structure of human Hsp40 (PDB entry: 3AGZ) was used as an initial structure for DNAJB1. The mutants DNAJB1$^{H244A}$, DNAJB1$^{E173A}$, DNAJB1$^{E174A}$ were constructed by a replacement of the amino acid histidine-244 by alanine, glutamic acid 173 or 174, respectively, using the CHARMM-GUI solvation builder[54–56]. For MD simulations proteins were embedded in a box of water of $12 \times 12 \times 12$ nm[57] and neutralized by sodium and chloride ions. All simulations were performed with the GROMACS 2018 version[19] and a timestep of 2 fs. Explicit water was represented by the TIP3P model[57,58], while the protein and ions were defined by the CHARMM36m force field[59]. The leap-frog algorithm was employed as an integrator and the LINCS algorithm[60] to constraint bonds connected to hydrogens atoms. Velocity rescaling[61], using a tau of 0.1 ps, was used for temperature coupling at 300 K and the Parrinello–Rahman algorithm for pressure coupling. The Verlet non-bonding cutoff scheme[62] for short-range interactions was set to 1.2 nm and the long-range interactions were described by particle mesh Ewald (PME). Systems were minimized using the steepest descent algorithm and further equilibrated by an NVT and NpT simulation applying a position restraint of 1000 kJ/mol to the heavy atoms of the protein for 0.1 ns, respectively. In a second set, an NpT equilibration of 1 ns with an unrestraint protein for all systems followed. Production runs were performed in an NVT ensemble for 500 ns each. Simulations were analyzed every 100 ps using the Visual Molecular Dynamics (VMD) software, the open-source community-developed PLUMED library version 2.6[63–65] and plots visualized by Python version 3.7.

## Model Generation of HTTExon1Q$_{48}$ via temperature replica exchange simulations

The HTTExon1Q$_{48}$ FASTA sequence of 113 amino acids length was used for a structure prediction based on sequence alignment by I-TASSER[24,25]. Results of the structure prediction are deposited on https://doi.org/10.5281/zenodo.6365426. Notably, this model only serves as a starting structure for the subsequently described refinement process of the peptide conformation by TIGER2h[26]. The resulting model (model 1) was solvated in a box of water and additional chloride ions added to neutralize the total system charge. Temperature replica exchange simulations were performed with the NAMD 3.0a9 version[66]. As for the protein, explicit water was represented by the TIP3P model[57,58], while the protein and ions were defined by the CHARMM36m force field[59]. Using hydrogen mass repartitioning with 3 amu hydrogen masses, the simulations were carried out with 4 fs time steps[67]. Periodic boundary conditions are applied to the system, and intermolecular interactions are considered up to 1 nm with a

switching function of 0.9 nm. Long-range electrostatic interactions are described by PME using a grid spacing of 0.1 nm[68,69]. The interaction pair list is generated every 20 steps for atoms within 1.6-nm distance. All covalent bonds are constraint with the SETTLE algorithm to the optimal length[70]. Temperature and pressure were maintained using Langevin thermo- and barostats[71,72]. The optimal parameters for replica exchange were set by the respective replica temperature and Antoine-equation (A=8.14019, B=1810.94 °C, C=244.485 °C, result in Torr)[73]. Replica exchange simulations were performed with the TIGER2h[26] method with 24 temperature replicas spanning 310–600 K. Temperatures for the replicas are assigned such that the temperature differences are logarithmically distributed over the temperature range. In the TIGER2h method, temperature replicas are cooled after a 16 ps sampling period to 310 K within 4 ps. While the protein was sampled in explicit solvent, the potential energies for determining exchanges using the Metropolis sampling criterion[74] were evaluated in implicit solvent (GB$_{OBC}$II) by calling OpenMM 8.4.5[75] on the protein structure after the cooling period. After evaluation of exchanges, dynamics were restarted with reassigned temperatures. Structural similarity between conformations from the TIGER2h simulations were investigated employing a dihedral principal component analysis (dPCA)[27]. Here, the protein backbone dihedral angles ϕ and ψ are transformed into cartesian coordinates and used as input parameters. dPCA has the advantage that the number of degrees of freedom are drastically reduced to the most important structural indicator. Additionally, the remaining degrees of freedom are translationally and rotationally invariant. Afterward, a coordinate pair consisting of the first and second principal component is assigned to each protein conformation. Together, the first and second principal components encode 25 and 10% of the total movement for the full sequence and fragment, respectively. This approach has been previously shown to define distinct protein conformations with reasonable accuracy. The resulting density is clustered with the density-based clustering algorithm DBSCAN[76] included in the scikit-learn library[77].

## DNAJB1-HTTExon1Q$_{48}$ complex formation by docking analysis and subsequent MD simulation

The HDOCK server[78–82] was used for docking of HTTExon1Q$_{48}$ to DNAJB1 variants. As an input structure for HTTExon1Q$_{48}$, a representative minimum energy structure from the most five prominent clusters were used as defined by dPCA, respectively. Equilibrated DNAJB1 structures of all studied variants were chosen from the previous MD simulations (from the last 50 ns of simulation). Additionally, the residues of the binding sites were provided to the program: 93–102 (P2) of HTTExon1Q$_{48}$ and 238–246 of DNAJB1. No more input parameters had to be supplied. Top ten predicted complex structures for every DNAJB1 variant and different clusters are stored under, https://doi.org/10.5281/zenodo.6365426, with respective prediction scores. In short, always the highest scored model was chosen, which predicted the binding sites to be in contact: DNAJB1WT-cluster1: model1, -cluster2: model1, -cluster5: model4; DNAJB1H244A-cluster1: model4, -cluster2: model3; DNAJB1E173A-cluster1: model14, -cluster2: model1; DNAJB1E174A-cluster1: model1, -cluster2: model1.

Further setup of the simulation cell for the DNAJB1-HTTExon1Q$_{48}$ complexes as well as minimization, equilibration and production runs have been performed as described under "Classical molecular dynamics simulation parameters". Afterward, 500 ns of simulation with a completely unrestrained system were executed.

The contact map implementation of the PLUMED software was used to either construct contact matrices or their reference matrices. For each residue studied, the geometric center of all atoms belonging to that amino acid were used as reference point for distance measurements. Distances were converted by a rational switching function, defining the contact distance (inflection point) at 1 nm, where a value of 1 represents a close contact and 0 no contact. Thereby, distances

become normalized and the values of the switching function unitless. Calculated distances were averaged over the last 400 ns of the MD simulation. Additionally, in the case of contact maps between HTTExon1Q$_{48}$ with DNAJB1 variants, averaging of interactions recorded for cluster 1 and 2 was performed to ease visual interpretation.

### Statistical analyses

Graphs and statistical analysis were performed using Microsoft Excel and GraphPad Prism 8 and 9 software. Data groups were compared via a one-way ANOVA. Tukey's and Dunnet's test were used for the statistical hypothesis testing, and the family-wise alpha threshold and confidence level was set at 0.05 (95% confidence level). The $P$ values of the data sets are provided in the Source data.

### Reporting summary

Further information on research design is available in the Nature Research Reporting Summary linked to this article.

## Data availability

Protein structures from the PDB shown used in this study are: 3AGZ, 3C7N, and 1NLT. The in silico data of this study like structure and trajectory files are made available under https://doi.org/10.5281/zenodo.6365426. PLUMED input files required to reproduce the results reported in this study are available on PLUMED-NEST (www.plumed-nest.org), the public repository of the PLUMED consortium as plumID: 22.012. The mass spectrometry proteomics data have been deposited to the ProteomeXchange Consortium via the PRIDE partner repository (http://www.proteomexchange.org) with the dataset identifier PXD031214. Source data are provided with this paper that contains the raw data of all graphs depicted in the main figures and Supplementary information.

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

## Acknowledgements

We acknowledge funding from the Deutsche Forschungsgemeinschaft (KI-1988/5-1) to J.K. We acknowledge the Cellular Imaging Core Facility at the FMP for their help and guidance in the live imaging experimental setup, and Claudia Rutz for her advice, generous material contribution and help in mammalian cell culture techniques. Computational resources were provided by the North-German Supercomputing Alliance (HLRN).

## Author contributions

A.M.S.M. and J.K. conceptualized the study, designed the experiments, and wrote the manuscript, with contributions from all other authors. A.M.S.M., M.L.P., Y.R., M.Ö., I.L.G., J.P., K.Z., M.K., M.E., J.V.V., S.K., F.L., and J.K. performed and analyzed the experiments and simulations. L.C.C., S.K., and J.K. provided suggestions for simulations, interpretation of the results, and supervision.

## Funding

## Competing interests

The authors declare no competing interests.
