## [Peer Review File · Nature Communications]

REVIEWER COMMENTS

Reviewer #1 (Remarks to the Author):

The manuscript by Ayala et al has compelling in vitro data to demonstrate novel findings on the mechanism of how DNAJB1 can suppress polyQ_{htt} aggregation.

However, the in vivo data to support this, is not very strong. Ayala et al does show that overexpression of wt DNAJB1, but not mt H244A DNAJB1, can suppress polyQ_{htt} aggregation.

What is not so convincing is, if this has relevance in cells. Therefore, as a minimum, the authors should be able to see increase in polyQ_{htt} aggregation in a cell line lacking DNAJB1 (a DNAJB1 KO cell line overexpressing polyQ_{htt}) and that this can be re-suppressed by re-introducing DNAJB1 into these KO cells – and that it cannot be rescued by re-introducing mt H244A DNAJB1.

Secondly it would be interesting to see how the H32Q mutation in the HPD motif of the J-domain suppresses polyQ_{htt} aggregation relative to the H244A mutation and relative to the double mutation (H32Q, H244A). I

In addition it would be good if the authors could repeat critical experiments using a H244Q mutation instead of the H244A, to show that the effect is not caused by the alanine 244 specifically.

Reviewer #2 (Remarks to the Author):

Ayala Mariscal et al. have characterized an interaction interface between chaperone complex DNAJB1/Hsc70/Agg2 and Huntingtin (HTT). A second proline-rich domain (PRD) of HTT was identified as a new binding site for chaperones and the hinge region between CTD I and CTD II domains of DNAJB1 was shown to bind PRD of soluble as well as aggregated HTT. Mutational studies point out the primary role of H244 of DNAJB1 in the disaggregation of HTT fibrils. The topic of the study as well as the experimental results are very interesting. However, I have several concerns regarding the computational part of this study, which are listed below.

MD simulations. First, something is wrong with Fig.4A bottom. There is no light gray, only gray, i.e., no results for DNAJB1 H244A. Also, it is not clear why some time intervals are missing and why the average

number of hydrogen bonds is either around 100 or around 200. Second, the authors conclude from the fibrillization assays that E174 “contributes to the activity of DNAJB1 in suppressing HTTExon1Q48 fibrilization with Hsc70 and Apg2” but there is no hypothesis explaining why, no simulations for this variant, and no structural analysis. This part of the work seems unfinished.

De novo modeling of HTTExon1Q4. Since the 3D structure of HTTExon1Q4 was obtained by modelling it would be helpful to share the used sequence in the Supplementary section. As there was no sequence provided, I modeled the N-terminal fragment of huntingtin (113 AAs) from Uniprot (<https://www.uniprot.org/uniprot/P42858.fasta>) using AlphaFold 2. Here is this fragment and I hope it is similar to the HTTExon1Q48:

```
MATLEKLMKAFESLKSFQQQQQQQQQQQQQQQQQQQQPPPPPPPPPPQLPQPPPQAQPLLQPQPPPPPPP  
PPGPAVAEELHRPKKELSATKKDRVNHCLTICENIVAQ
```

Five top-ranking predictions are illustrated in the attachment.

There are two common features in these structures:

1. N17 and PolyQ as well as the C-terminus form alpha-helices;
2. The middle section forms a turn to provide anti-parallel contacts for the terminal helices.

The prediction for the middle section (containing both proline-rich regions) is common for the intrinsically disordered regions (IDRs). Presented structures are good starting models for the MD for someone who is interested in the ways they will collapse to organize compact globules. Then, if the sampling is sufficient, using clustering techniques, one can establish the number of representative structures. The authors chose another approach and calculated five different but energetically almost equal conformational states. The differences between the theoretically predicted structures and the absence of experimental validation raise the question: why the first cluster was chosen as a representative structure used in molecular docking? According to the energy estimations, all five calculated conformations will be noticeably populated in the HTTExon1Q48 assembly at room temperature and the first cluster structure will represent only the fifth part of the assembly. Summing up, that part of the manuscript provides a weak foundation for the later molecular docking analysis.

Molecular docking. The molecular docking part of the research is also very questionable. First, the “Materials and methods” section is not sufficient to understand what was done. How was the repositioning chosen? What was the sampling grid? How many poses were calculated? What was the score distribution? The statements “the default docking parameters” as a setup and “a -13.3 kcal/mol binding free energy” as a result are not enough. Generally, protein-protein docking is not the most robust method, especially in its rigid form, so the results should be carefully assessed using all available means.

Overall, it is not quite clear what these computational results bring to the study. They seem to be strongly affected by the previously obtained experimental data and at the same time are claimed to be confirming and validating them (e.g. statements such as “molecular dynamics simulation and docking analyses confirmed the experimentally identified binding interface” and “In silico DNAJB1-HTTExon1Q48

complex validates experimentally observed contact sites”). The mentioned above issues should be addressed if the authors believe that the computational part is essential for the study.

Reviewer #3 (Remarks to the Author):

The authors report the analysis of the interaction between DNAJB1 and the polyQ-adjacent proline rich domain (PRD) of HTT. Study of this interaction is key to understanding the molecular mechanisms that allow DNAJB1 to prevent or reverse HTT accumulations.

The results are driven by a cross-linker/mass-spec analysis that detail the likely architecture of the DNAJB1/HSC70/HTT interaction. These data will be of significance in the field.

The authors use point mutations to confirm the mass-spec results, showing indeed that the identified region (242-244) plays a role in the inhibition of fibril formation, but does impact DNAJB1 structurally (DSF/CD/SEC). These results were shown to be specific to HTT and the likely role of these mutations analysed by molecular dynamics.

I can find no clear flaws in the experimental design or analysis and the methodology is detailed and composed of a wide variety of techniques. I would recommend publication after (very) minor corrections.

1. could the authors more comment on the potential for these mutations to impact Hsc70 binding? The active site is distal, but is there any evidence for changes due to mutations in the molecular dynamics?
2. figure 6 SI2c - please quote errors on expt (rather than calculated?) measurements

Reviewer #4 (Remarks to the Author):

Ayala Mariscal et al. report on new regions in DNAJB1 that interact with HTT near the polyQ stretch. They demonstrate the importance of these regions to the prevention of aggregation of HTT, but not other substrates, by the DNAJB1 system. My understanding from the text is that the study was first informed by XL-MS results, although it is almost entirely based on other biochemical assays. I am not an expert on HTT or DNAJB1, but I think that these are certainly noteworthy findings especially for the research on HTT aggregation.

Nearly all conclusions are well based on the results, and I somewhat disagreed with only one: The authors show in Figure 3 that PRD2 is central for the prevention of HTT aggregation. They focused on PRD2 and not PRD1, because the cross-link that they have identified occurred on PRD2. However, since the XL-MS data are very sparse, I would not set too much store on them (see below). It may very well be that PRD1 is also interacting with the HBM, but that the conditions for identifying a cross-link around PRD1 are prohibitive (lack of elastase cut sites, etc.). I therefore urge the authors to repeat the experiments of Figure 3 with a PRD1-deletion in order to get a more complete picture of the interaction from the HTT side.

Regarding the XL-MS presentation. I am fully aware that the HTT sequence is extremely challenging for XL-MS both in availability of reactive sidechains and protease cut sites. Therefore, the choices of a semi-specific cross-linker and an unusual protease (elastase) are well justified. Accordingly, I am not surprised that only a few cross-links were identified. However, the presentation of the XL-MS data is very partial, and I ask the authors to complete it as follows:

- 1) Provide a complete and explicit list of all the cross-links that were identified (From the current text I am uniformed whether: Are there only two? Are some cross-link sites represented by more than one peptide pair?)
- 2) Provide annotated MS/MS spectra for all the identified cross-links. Since we are dealing with a very small number of cross-links, that should not take up too much space.
- 3) Make sure that the MS/MS annotation is complete. The MS/MS spectrum in Figure S1 seems OK (XL-MS-wise), but the fragments of Peptide A are not marked.

Minor points:

1. The insert in Fig. 4a is completely meaningless as is. Either it is removed, or made to zoom-in on the 95-100 region in the Y-axis.
2. There is huge variability in the results presented in the ATPase assays (Fig. S2b) between the different mutants. For example, the H244R mutant is x2 higher than the WT. Is this physiological, or the 'noise' of the assay? I wonder if the authors could comment on this in the text.

Rebuttal letter for:

Identification of a HTT-specific binding motif in DNAJB1 essential for suppression and disaggregation of HTT

by Ayala et al.

We appreciate the positive and constructive feedback of all four reviewers and provide our point-to-point response below. You will find our response in red below each comment of the reviewers.

In summary, we have performed the following experiments to address the comments of all reviewers:

- Analysis of DNAJB1^{H32Q} in FRET, ATPase and luciferase refolding assays
- Analysis of DNAJB1^{H32Q/H244A} in FRET, ATPase and luciferase refolding assays
- Analysis of DNAJB1^{H244Q} in FRET, ATPase and luciferase refolding assays
- Analysis of the suppression of aggregation of HTTExon1Q₄₈ΔP1 by the trimeric chaperone complex
- Analysis of the role of *dnajb1* depletion on the aggregation of HTTExon1Q₉₇ in HEK cells + rescue upon overexpression of DNAJB1^{wt} vs DNAJB1^{H244A}
- MD simulations of several HTTExon1Q₄₈ cluster models in complex with DNAJB1 variants (wt and three selected mutants) as well as an analysis of the mutual contact maps.

We also provided the raw data and deposited them on the PRIDE repository for:

- XLMS
- *in situ* studies

Any changes in the text of our revised manuscript are highlighted in yellow.

REVIEWER COMMENTS

Reviewer #1 (Remarks to the Author):

The manuscript by Ayala et al has compelling in vitro data to demonstrate novel findings on the mechanism of how DNAJB1 can suppress polyQ^{htt} aggregation. However, the in vivo data to support this, is not very strong. Ayala et al does show that overexpression of wt DNAJB1, but not mt H244A DNAJA1, can suppress polyQ^{htt} aggregation. What is not so convincing is, if this has relevance in cells. Therefore, as a minimum, the authors should be able to see increase in polyQ^{htt} aggregation in a cell line lacking DNAJB1 (a DNAJB1 KO cell line overexpressing polyQ^{htt}) and that this can be re-suppressed by re-introducing DNAJB1 into these KO cells – and that it cannot be rescued by re-introducing mt H244A DNAJB1.

We thank the reviewer for the positive assessment of our study and the constructive criticism. Specifically, the reviewer asks about the relevance of DNAJB1 in HTTpolyQ-expressing cells. We could previously show that depletion of *dnajb1* by siRNA led to an increased aggregation of the endogenous HTT (Q₄₄) in Huntington's disease patient-derived neuronal progenitor cells (Scior et

al., 2018). In the same study, we could demonstrate that overexpression of DNAJB1 could rescue the aggregation of HTTExon1Q₉₇ in HEK293T cells (Scior et al., 2018).

In this manuscript, we build on this rescue experiment and demonstrated that the H244A mutant of DNAJB1 failed to rescue the aggregation of HTTExon1Q₉₇ upon overexpression (Fig. 9a-c).

The reviewer suggests an additional experiment to deplete *dnajb1* in HTTpolyQ expressing cells that should result in an increase in HTTpolyQ aggregation and to transfect the cells again to overexpress either DNAJB1^{wt} or DNJB1^{H244A} to test if the aggregation of HTTpolyQ can be rescued by DNAJB1^{wt}, but not by DNAJB1^{H244A}.

We performed this experiment (Figs. 9d+e) and could show that depletion of *dnajb1* by siRNA led to an enhanced aggregation of HTTExon1Q₉₇ in HEK cells. This aggregation phenotype could be rescued by overexpressing DNAJB1^{wt} but not by DNAJB1^{H244A}.

Secondly it would be interesting to see how the H32Q mutation in the HPD motif of the J-domain suppresses polyQhtt aggregation relative to the H244A mutation and relative to the double mutation (H32Q, H244A).

The reviewer asks to test the H32Q mutant of DNAJB1 in the FRET assay to assess if the mode of action of DNAJB1 is dependent or independent of Hsc70. The HPD motif of J-domain proteins is required to induce the ATPase of Hsc70 by the partner J-domain protein. A mutation of the histidine residue disrupts the cooperation between the J-domain protein and Hsc70. We hence introduced the H32Q mutation into DNAJB1 (DNAJB1^{H32Q}) and tested its ability to suppress HTTExon1Q₄₈ aggregation together with Hsc70 and Apg2. DNAJB1^{H32Q} fails to suppress HTTExon1Q₄₈ aggregation together with Hsc70 and Apg2 (Supplemental Figure 6c). These data further support our previous observation that all three chaperones: DNAJB1, Hsc70 and APG2 are required to suppress the aggregation of HTTExon1Q₄₈ (Scior et al., 2018). The double mutant, DNAJB1^{H32Q/H244A} is as expected also inactive in the suppression of HTTExon1Q₄₈ fibrilization together with Hsc70 and Apg2 (Supplemental Figure 6c).

In addition it would be good if the authors could repeat critical experiments using a H244Q mutation instead of the H244A, to show that the effect is not caused by the alanine 244 specifically.

The reviewer asks to substitute H244 also by glutamine (Q). We would like to point out that we have already replaced H244 by multiple aa in the original manuscript: We have generated H244A, H244R, H244F mutants and could demonstrate that H244A and H244F behaved similarly and were essentially inactive towards suppression of HTTExon1Q₄₈ aggregation together with Hsc70 and Apg2 (Fig. 1d), whereas H244R showed an intermediate activity (Fig. 1d). In the revised manuscript we have now also generated a DNAJB1^{H244Q} mutant as requested and observed also an intermediate activity in suppressing HTTExon1Q₄₈ aggregation together with Hsc70 and Apg2 (Fig. 1d). Glutamine is a non-charged, yet polar aa that is able to form hydrogen bonds and thus stabilizes the binding site for HTT similarly as arginine, as explained with the help of novel Molecular Dynamics simulations (see answer to Reviewer #2).

Reviewer #2 (Remarks to the Author):

Ayala Mariscal et al. have characterized an interaction interface between chaperone complex DNAJB1/Hsc70/Apg2 and Huntingtin (HTT). A second proline-rich domain (PRD) of HTT was identified as a new binding site for chaperones and the hinge region between CTDI and CTDII domains of DNAJB1 was shown to bind PRD of soluble as well as aggregated HTT. Mutational studies point out the primary role of H244 of DNAJB1 in the disaggregation of HTT fibrils. The topic of the study as well as the experimental results are very interesting. However, I have several concerns regarding the computational part of this study, which are listed below.

We thank the reviewer for the constructive criticism and have now performed several new computational analyses that explain more in detail the role of H244 in stabilizing DNAJB1/HTT complexes. We have also rewritten most of the relevant text and modified the corresponding figures.

MD simulations. First, something is wrong with Fig.4A bottom. There is no light gray, only gray, i.e., no results for DNAJB1 H244A.

We agree with the reviewer that there were no results shown for the DNAJB1^{H244A}, as no hydrogen bonds exist between E173 and A244. We have now modified the figure in a way that it shows only the distance between atoms that are involved in hydrogen bonding of residues E173 and H244 in DNAJB1^{WT} to avoid confusion (now Supplemental Figure 4a).

Also, it is not clear why some time intervals are missing and why the average number of hydrogen bonds is either around 100 or around 200.

We thank the reviewer for noticing this mistake. Instead of “100” and “200”, the labels should have been “1” and “2”. As mentioned above, we have now modified the figure to better convey both the original point and the new knowledge generated by additional simulations (see below).

Second, the authors conclude from the fibrillization assays that E174 “contributes to the activity of DNAJB1 in suppressing HTTExon1Q48 fibrilization with Hsc70 and Apg2” but there is no hypothesis explaining why, no simulations for this variant, and no structural analysis. This part of the work seems unfinished.

We agree with the reviewer and have now added *in silico* data for DNAJB1^{H244A}, DNAJB1^{E173A} and DNAJB1^{E174A} (Figs. 4a+c and Supplemental figures 3 and 4). In particular, these mutants have been first simulated in separate MD runs alone (in dimer form) to achieve equilibrium structures. Second, we used these structures to perform docking analyses with all five cluster structures of HTTExon1Q₄₈ (two of which resulted in stable complexes). A whole new section was added to the manuscript, describing the atomistic details of the HBM of the different DNAJB1 variants in contact with the P2 of HTTExon1Q₄₈. In particular, the role of H244 in stabilizing complexes with HTT is

explicitly explained by the simulations, and the role of E174 in providing a stable binding contact to Hsc70 is inferred indirectly by the simulation results and existing literature knowledge.

De novo modeling of HTTExon1Q4. Since the 3D structure of HTTExon1Q4 was obtained by modelling it would be helpful to share the used sequence in the Supplementary section. As there was no sequence provided, I modeled the N-terminal fragment of huntingtin (113 AAs) from Uniprot (<https://www.uniprot.org/uniprot/P42858.fasta>) using AlphaFold 2. Here is this fragment and I hope it is similar to the HTTExon1Q48:

```
MATLEKLMKAFESLKSFQQQQQQQQQQQQQQQQQQQQQQPPPPPPPPPPQLPQPPPQAQPLL  
PQPQPPPPPPPPPPGPAVAEEPLHRPKKELSATKKDRVNHCLTICENIVAQ
```

The correct aa sequence of HttExon1Q₄₈ differs in the number of glutamines as well as in the sequence of the PRD domain, and is:

```
MATLEKLMKAFESLKSFQQQQQQQQQQQQQQQQQQQQQQQQQQQQQQQQQQQQQQQQQQQQQ  
QQQQQPPPPPPPPPPQLPQPPPQAQPLLQPPPPPPPPPPGPAAAEEPLHRP
```

We used a protein with this sequence for the *in silico* work as well as for all biochemical assays. We have added the sequence of HTTExon1Q₄₈ to the Materials and Method section in the revised manuscript.

Five top-ranking predictions are illustrated in the attachment.

There are two common features in these structures:

1. N17 and PolyQ as well as the C-terminus form alpha-helices;
2. The middle section forms a turn to provide anti-parallel contacts for the terminal helices.

The prediction for the middle section (containing both proline-rich regions) is common for the intrinsically disordered regions (IDRs). Presented structures are good starting models for the MD for someone who is interested in the ways they will collapse to organize compact globules. Then, if the sampling is sufficient, using clustering techniques, one can establish the number of representative structures. The authors chose another approach and calculated five different but energetically almost equal conformational states.

First of all, we would like to acknowledge the effort of the reviewer to model the HTTExon1 via AlphaFold2. Unfortunately, the used sequence only harbors 21 Qs and not 48, as we are using in our study. Nevertheless, structural features like the alpha-helices in the N17 and poly-Q domains are similar to our predicted models (especially cluster 2 -5), as well as the PRD domains, which fold into unstructured coils.

We would further like to point out that the structures we have obtained through the i-Tasser modelling (basically equivalent to AlphaFold2) are also only starting models for our enhanced-sampling MD method TIGER2h. This method allows for a much more comprehensive sampling of the configurational phase space of the protein model than standard MD runs. In particular, the sampling of the method is independent of the starting structure, although educated guesses as starting points improve the convergence efficiency. Therefore, using predictions generated by either i-Tasser or AlphaFold2 does not significantly influence the final outcome. Details on the structural prediction capability and limitations of TIGER2h simulations are described in two original papers:

1. Kulke, M., Geist, N., Möller, D. & Langel, W. Replica-Based Protein Structure Sampling Methods: Compromising between Explicit and Implicit Solvents. *J Phys Chem B* 122, 7295–7307 (2018).

2. Geist, N., Kulke, M., Schulig, L., Link, A. & Langel, W. Replica-Based Protein Structure Sampling Methods II: Advanced Hybrid Solvent TIGER2hs. *J Phys Chem B* 123, 5995–6006 (2019).

In the revised version of the manuscript, we now explicitly show the convergence of our simulations in Supplemental Figure 3c.

TIGER2h is a replica-exchange method, in which the peptide conformations are exchanged between different replicas, where higher replicas are simulated at a higher temperature and therefore behave more ergodically than e.g. the ground replica at 300K. By this, we ensure a sufficient sampling of the peptide's phase space to explore as many possible conformations as possible. Subsequently, dPCA was used to cluster the conformers, using the unbiased ground replica (we assume that the reviewer was referring to similar approaches when mentioning "clustering techniques"). Under the assumption of ergodicity, the probability of clusters present in the ground (unbiased) replica can be converted to a free-energy map. In our case, five prominent clusters were found, where cluster 2 has the lowest free energy and therefore a high probability to occur in the ground replica, followed by cluster 1, 3, 5 and 4. This is of course reflected by the respective number of microstates included in each cluster.

In summary, we do believe that our approach validly delivers representative structures of HTTExon1Q₄₈, and does not differ in essence from the workflow suggested by the reviewer. On the contrary, the use of enhanced-sampling methods rather than standard MD ensures sufficient phase-space sampling, as indicated by the now presented convergence analysis.

The differences between the theoretically predicted structures and the absence of experimental validation raise the question: why the first cluster was chosen as a representative structure used in molecular docking?

In the original manuscript, we focused only on one cluster (numbered 1), which contained 20% of our identified conformers and presented a compact and globular structure. However, we agree with the reviewer that this is an arbitrary choice. In the revised manuscript, we have thus considered all five clusters for docking simulations (followed by standard MD runs) to test their ability to form stable complexes to DNAJB1 (Fig. 4c).

According to the energy estimations, all five calculated conformations will be noticeably populated in the HTTExon1Q₄₈ assembly at room temperature and the first cluster structure will represent only the fifth part of the assembly. Summing up, that part of the manuscript provides a weak foundation for the later molecular docking analysis.

As mentioned above, we have now included the five most populated clusters in the docking analysis. Thereby, we already account for almost 70% of the conformers that were identified by our enhanced sampling MD approach. However, we note that clusters 3 and 4 could not be docked by the employed docking server HDock with the predefined interfaces of HBM and P2. This can be

explained by a masked/shielded P2 domain by the N17 and polyQ domain (Supplemental figure 3b).

Molecular docking. The molecular docking part of the research is also very questionable. First, the “Materials and methods” section is not sufficient to understand what was done. How was the prepositioning chosen? What was the sampling grid? How many poses were calculated? What was the score distribution? The statements “the default docking parameters” as a setup and “a -13.3 kcal/mol binding free energy” as a result are not enough. Generally, protein-protein docking is not the most robust method, especially in its rigid form, so the results should be carefully assessed using all available means.

We thank the reviewer for pointing out the deficiencies of the used docking method and the missing parameters. After careful assessment, we decided to change from the AutoDock4 method, which is clearly defined to dock small ligands to a protein, to the HDock server (Yan, Y., Tao, H., He, J. *et al.* The HDock server for integrated protein–protein docking. *Nat Protoc* **15**, 1829–1852 (2020). <https://doi.org/10.1038/s41596-020-0312-x>).

HDock is still a rigid-body docking algorithm (and thus necessitates follow-up MD simulations, as performed here), but designed for the case that we are studying, namely protein-protein complexes. We agree that a rigid docking algorithm eliminates a lot of important degrees of freedom to the proteins that are docked and more advanced methods like SwarmDock (Moal IH, Chaleil RAG, Bates PA. Flexible Protein-Protein Docking with SwarmDock. *Methods Mol Biol.* 2018;1764:413-428. doi: 10.1007/978-1-4939-7759-8_27. PMID: 29605931.) are currently developed to tackle these difficulties. However, we extensively sampled the HTTExon1Q₄₈ conformations in advance by TIGER2h and used the five most prominent clusters for docking in the revised manuscript. Additionally, subsequent MD simulations of the docked complexes allowed for the relaxation and equilibration of the structures, where the docking algorithm only served to generate initial starting conformations of putative DNAJB1/HTTExon1Q₄₈ complexes (both for the wt and three relevant mutations). One could in principle perform (very expensive) enhanced-sampling MD simulations also of the derived complexes to fully take into account the e.g. high flexibility of the P2 domain in the intrinsically disordered HTTExon1Q₄₈. However, since the experiments deliver compelling evidence for the predicted binding interfaces of DNAJB1 to HTTExon1Q₄₈, we concluded that plain MD simulations of 500 ns already provide a good basis to understand the relevant protein-peptide interactions. A more extensive study about how DNAJB1 might undergo conformational changes upon HTTExon1Q₄₈ and how large the phase space of the docked HTTExon1Q₄₈ peptide is, should be a subject of further, more advanced computational studies, where one should also go beyond the use of flexible docking algorithms. Clearly, structural information is also still lacking; e.g. whether DNAJB1 binds HTTExon1Q₄₈ in its monomeric or rather dimeric form, and whether two HTTExon1Q₄₈ moieties may bind simultaneously to the DNAJB1 dimer. These limitations of the used *in silico* investigation method are reported in the discussion section of the revised manuscript.

Overall, it is not quite clear what these computational results bring to the study. They seem to be strongly affected by the previously obtained experimental data and at the same time are claimed to be confirming and validating them (e.g. statements such as “molecular dynamics simulation and docking analyses confirmed the experimentally identified binding interface” and “In silico DNAJB1-HTTExon1Q₄₈ complex validates experimentally observed contact sites”). The mentioned above

issues should be addressed if the authors believe that the computational part is essential for the study.

We agree with the reviewer that our simulation approach is strongly based upon the obtained experimental data. We want to point out that this is intended, as the docking procedure should not validate the experimentally identified binding sites, but only reveal the atomistic details of such interactions upon protein-peptide binding.

The additional simulations and analyses now are in a much better position than in the original manuscript, and they include explicit consideration of the interfaces between HTT and DNAJB1^{wt}, DNAJB1^{H244A}, and DNAJB1^{E173A}. It is true that the simulations still do not “confirm” or “validate” the experimental observations. Rather, they deliver details of the mutual protein-peptide interactions that rationally explain the key role of H244. They also explain why mutations with hydrophobic residues (H244F, H244A) cause a complete loss of activity, whereas mutations with hydrophilic residues (H244R, H244Q) result in an intermediate activity between DNAJB1^{wt} and, for instance, DNAJB1^{H244A}.

We have now modified the statements mentioned by the reviewer in the revised manuscript. In particular, the following sentence has been added: “In addition, molecular dynamics simulation revealed atomistic details about the binding of HTTExon1Q₄₈, which can occur through the backbone atoms of H244 or side chain atoms in the case of DNAJB1^{wt} and DNAJB1^{E173A}, respectively. DNAJB1^{H244A} showed no complex formation with different HTTExon1Q₄₈ conformers, confirming the experimental observables that this mutant cannot prevent HTT fibrilization.”.

In summary, we believe that the computational results now provide the following insights, which are complementary to all experimental techniques used in this study:

1. Molecular (atomic-scale) details of the hinge region between CTDI and CTDII of DNAJB1, characterizing intra-protein interactions of certain amino acids like H244 with E173 or E174 with S171, C179 and K181. Only due to this analysis, mutation variants DNAJB1^{E173A} & DNAJB1^{E174A} have been constructed and tested *in vitro* in the HTTExon1Q₄₈ fibrilization assays (Fig. 4b).
2. A set of possible three-dimensional structures of HTTExon1Q₄₈, constructed by the enhanced sampling technique TIGER2h. As no experimental structure is currently available for this peptide, our approach is able to provide converged estimates of the conformational phase space that can be explored by the algorithm, within the limits of force field accuracy. Ergodicity was taken care of by extensive replica exchanges between different temperature replicas (Supplemental Figure 3).
3. Atomistic structures of DNAJB1/HTTExon1Q₄₈ complexes for DNAJB1^{wt} and different mutation variants. Only due to the atomic-level scale of MD simulations, interactions that are displayed in figure 4c can be assessed and made visible. We were able to pinpoint the specific interactions of residue 244 with HTTExon1Q₄₈ and also show that residues 173 and 174 are not in direct contact with the peptide, however influencing its folding capability indirectly.

Reviewer #3 (Remarks to the Author):

The authors report the analysis of the interaction between DNAJB1 and the polyQ-adjacent proline rich domain (PRD) of HTT. Study of this interaction is key to understanding the molecular mechanisms that allow DNAJB1 to prevent or reverse HTT accumulations.

The results are driven by a cross-linker/mass-spec analysis that detail the likely architecture of the DNAJB1/HSC70/HTT interaction. These data will be of significance in the field.

The authors use point mutations to confirm the mass-spec results, showing indeed that the identified region (242-244) plays a role in the inhibition of fibril formation, but does impact DNAJB1 structurally (DSF/CD/SEC). These results were shown to be specific to HTT and the likely role of these mutations analysed by molecular dynamics.

I can find no clear flaws in the experimental design or analysis and the methodology is detailed and composed of a wide variety of techniques. I would recommend publication after (very) minor corrections.

1. could the authors more comment on the potential for these mutations to impact Hsc70 binding? The active site is distal, but is there any evidence for changes due to mutations in the molecular dynamics?

We thank the reviewer for the very positive assessment of our work.

This reviewer asks to comment on the potential effect of the H244 mutation of DNAJB1 on Hsc70 binding. We could show that DNAJB1^{H244A} is still able to induce the ATPase activity of Hsc70 and supports other chaperone activities such as suppression of A β ₁₋₄₂ fibrilization and refolding of denatured luciferase (Figs. 2b-c and Supplemental Fig. 2b). Thus, it does not seem that the H244A affects the interaction with Hsc70. And indeed, the binding interface of DNAJB1 and Hsc70 does not overlap with the HTT binding motif (Fig. 7c).

2. figure 6 SI2c - please quote errors on expt (rather than calculated?) measurements

We thank the reviewer for pointing out this mistake and we have corrected it accordingly.

Reviewer #4 (Remarks to the Author):

Ayala Mariscal et al. report on new regions in DNAJB1 that interact with HTT near the polyQ stretch. They demonstrate the importance of these regions to the prevention of aggregation of HTT, but not other substrates, by the DNAJB1 system. My understanding from the text is that the study was first informed by XL-MS results, although it is almost entirely based on other biochemical assays. I am not an expert on HTT or DNAJB1, but I think that these are certainly noteworthy findings especially for the research on HTT aggregation.

Nearly all conclusions are well based on the results, and I somewhat disagreed with only one: The authors show in Figure 3 that PRD2 is central for the prevention of HTT aggregation. They focused on PRD2 and not PRD1, because the cross-link that they have identified occurred on PRD2. However, since the XL-MS data are very sparse, I would not set too much store on them (see below). It may very well be that PRD1 is also interacting with the HBM, but that the conditions for identifying a cross-link around PRD1 are prohibitive (lack of elastase cut sites, etc.). I therefore

urge the authors to repeat the experiments of Figure 3 with a PRD1-deletion in order to get a more complete picture of the interaction from the HTT side.

We thank the reviewer for the positive assessment of our work and for pointing out that the interaction of DNAJB1 might not be limited to the P2 site and might expand to the whole PRD. To address this point, we have analyzed the ability of the chaperones to suppress the aggregation of HTTExon1Q₄₈ lacking the P1 domain (HTTExon1Q₄₈ΔP1). We could previously demonstrate that the first proline stretch of the PRD affects the aggregation propensity of HTTExon1 independently of chaperones. Deletion of the first proline stretch delays the aggregation (Pigazzini et al., 2021). Hence, we performed the FRET fibrilization assay for an expanded time to ensure the formation of amyloid fibrils of HTTExon1Q₄₈ΔP1 as reflected by the typical sigmoidal fibrilization curve (Fig. 3c). The T1/2 time of the aggregation of HTTExon1Q₄₈ΔP1 is 26 h compared to 10 h for HTTExon1Q₄₈ (Fig 3c). Notably, although the chaperone complex could not fully suppress the aggregation, it could significantly delay the onset of HTTExon1Q₄₈ΔP1 aggregation to a T1/2 of 35 h (Fig. 3c). We conclude that the chaperones are still able to bind and interact with HTTExon1Q₄₈ΔP1. There are a number of reasons for the absence of a complete suppression: 1st the deletion of the P1 could affect the conformation and hence aggregation kinetics of the protein independently of the chaperones as we have indeed observed (Pigazzini et al., 2021). 2nd a potential conformational change might also affect the accessibility of the P2 for binding by the chaperones and 3rd the P1 site might not be an initial contact site, but could be bound by the chaperones during their folding cycle.

Regarding the XL-MS presentation. I am fully aware that the HTT sequence is extremely challenging for XL-MS both in availability of reactive sidechains and protease cut sites. Therefore, the choices of a semi-specific cross-linker and an unusual protease (elastase) are well justified. Accordingly, I am not surprised that only a few cross-links were identified. However, the presentation of the XL-MS data is very partial, and I ask the authors to complete it as follows: 1) Provide a complete and explicit list of all the cross-links that were identified (From the current text I am uniformed whether: Are there only two? Are some cross-link sites represented by more than one peptide pair?)

We detected eleven crosslinks in the DNAJB1-HTTExon1Q₄₈ crosslinking experiment and 46 crosslinks in the HTTExon1Q₂₃-HSC70 crosslinking experiment. Most crosslinks are represented by one peptide pair, but multiple CSMs (crosslink spectrum matches). This information is shown in the 'CSM' column in the crosslink result file.

The data are now available via ProteomeXchange with identifier PXD031214

Project Name: Identification of a HTT-specific binding motif in DNAJB1 essential for suppression and disaggregation of HTT

Username: reviewer_pxd031214@ebi.ac.uk

Password: KTfXPPZT

2) Provide annotated MS/MS spectra for all the identified cross-links. Since we are dealing with a very small number of cross-links, that should not take up too much space.

See above. All annotated spectra of crosslinks are deposited in All_crosslinks_spectra_merged.pdf.

3) Make sure that the MS/MS annotation is complete. The MS/MS spectrum in Figure S1 seems OK (XL-MS-wise), but the fragments of Peptide A are not marked.

We used SDA crosslinker in all crosslinking experiments. SDA has one end (NHS ester) that reacts towards lysine residues and the other side (diazirine) is unspecific. We labeled all fragments from the Lys-cross-linked peptide, but for the other peptide we did not label the fragments as we could not pinpoint which amino acid is crosslinked. We therefore considered the crosslinking site of the diazirine can be on any amino acids of the crosslinked peptide.

Minor points:

1. The insert in Fig. 4a is completely meaningless as is. Either it is removed, or made to zoom-in on the 95-100 region in the Y-axis.

We thank the reviewer for pointing out irregularities with this figure. We have changed the figure completely in the revised manuscript.

2. There is huge variability in the results presented in the ATPase assays (Fig. S2b) between the different mutants. For example, the H244R mutant is x2 higher than the WT. Is this physiological, or the 'noise' of the assay? I wonder if the authors could comment on this in the text.

The ATPase activity reflects on the activation of the intrinsic ATPase activity of Hsc70 by the different DNAJB1 variants. There is a certain extend of variation as indicated by the error bars for some protein combinations. Higher ATPase rates could be explained by stronger interactions between Hsc70 and the respective J-protein. We commented on the observed differences between the DNAJB1 variants on the ATPase activation of Hsc70 in the figure legends of Supplemental Figure 2b. It reads as follows: "The extend of ATPase activation of Hsc70 by the different DNAJB1 variants could be due to altered affinities to Hsc70."

REVIEWERS' COMMENTS

Reviewer #1 (Remarks to the Author):

The authors have adressed my questions to a satisfactory extend. I therefor recommend that the journal publish this study

Reviewer #2 (Remarks to the Author):

The authors have performed additional computational studies and modified the corresponding part of the manuscript. I thank the authors for their detailed responses and for addressing the concerns I had with the previous version. I wish them good luck in their further research.